

# Does the Asian Summer Monsoon Play a Role in the Stratospheric Aerosol Budget of the Arctic?

Sandra Graßl[1,2], Christoph Ritter[1], Ines Tritscher[3], and Bärbel Vogel[3]

[1]Alfred Wegener Institute, Helmholtz Centre for Polar and Marine Research, 14473 Potsdam, Germany
[2]Institute of Physics and Astronomy, University of Potsdam, Karl-Liebknecht 24/25, 14476 Potsdam, Germany
[3]Forschungszentrum Jülich, Institute of Energy and Climate Research – Stratosphere (IEK-7), Jülich, Germany

**Correspondence:** Sandra Graßl (Sandra.Grassl@awi.de)

**Abstract.** The southeast Asian monsoon has a strong convectional component, with which aerosols are able to be lifted up into the lower stratosphere. Due to usually long lifetimes and long-range transport aerosols remain there much longer than in the troposphere and are also able to be advected around the globe. Our aim of this study is a synergy between modelled tropical aerosol tracers by Chemical Lagrangian Model of the Stratosphere (CLaMS) and KARL (Koldewey Aerosol Raman

Lidar) at AWIPEV, Ny-Ålesund in the Arctic, by comparing back- and forward trajectories with exemplary days of Lidar measurements as well as analyse the stratospheric aerosol background. We use global 3-dimensional Lagrangian transport simulations including surface origin tracers as well as back-trajectories to identify source regions of the aerosol particles measured over Ny-Ålesund. We analysed Lidar data for the year 2021 and found the stratosphere generally clear, without obvious aerosol layers from volcanic eruptions or forest fires. Still an obvious annual cycle of the backscatter coefficient

with higher values in late summer to autumn and lower values in late winter have been found. Results from CLaMS model simulations indicate that from late summer to early autumn filaments with high fractions of air which originate in South Asia – one of the most polluted regions in the world – reach the Arctic in altitudes between 360 K and 380 K potential temperature. We found a coinciding measurement between the overpass of such a filament and Lidar observations, we estimated that backscatter and depolarisation increased by roughly 15% during this event compared to the background aerosol concentration. Hence we

demonstrate that the Asian summer monsoon is a weak but measurable source for Arctic stratospheric aerosol in late summer to early autumn.

## 1 Introduction

Already in 1960 Junge et al. (1961a) characterised some basic properties of stratospheric aerosols like size distribution, physical structure and chemical composition up to 30 km altitude. Stratospheric aerosol can either be long-range transported or

even directly created in the stratosphere and has a high sulfuric content (Junge et al., 1961a; Kremser et al., 2016), they have a big impact on the local ozone layer but also on the radiative budget with a cooling factor due to additional scattering of direct solar radiation. Already Junge et al. (1961a) pointed out different mechanisms acting on stratospheric aerosols, like horizontal mixing due to large-scale exchange by intrusions through the tropopause, meridional, hemispheric circulations or gravitational settling of particles determined by Stokes' law. Junge et al. (1961a) as well as Murphy et al. (2021) found in stratospheric



in-situ measurements a remarkable constant layer of particles in the accumulation mode (from 0.1 $\mu$m to 1 $\mu$m). Particles larger than that are created by oxidation of $H_2S$ and $SO_2$ in the layer they are found. Strong zonal winds in the stratosphere lead to a fast homogenisation of aerosols, while vertical and meridional transport is controlled by the Brewer-Dobson-Circulation (BDC) (Holton et al., 1995; Butchart, 2014). The BDC is characterised by a meridional transport from rising air at the tropics towards high latitudes and a quasi-horizontal mixing over the extra-tropics (McIntyre and Palmer, 1983). This circulation and

mixing controls not only long-range transport (Shepherd, 2002) but also ageing of aerosols (Waugh and Hall, 2002).

As Junge et al. (1961a) found out, life-times of aerosols in the lower stratosphere are in the order of magnitude of several hundred days and can easily be advected to higher latitudes and distributed over the entire globe. The absence of water vapour and removal by precipitation does not take place in the stratosphere. Aerosols are usually removed by sedimentation.

During the first four years of CALIPSO measurements (Cloud-Aerosol Lidar and Infrared Pathfinder Satellite Observations)

an Asian Tropopause Aerosol Layer (ATAL) was found during the Asian summer monsoon season in altitudes of 13 to 18 km equivalent to 360 K to 420 K potential temperature, while the Junge layer is usually at around 20 km (Junge et al., 1961b). With usually occurring depolarisation values of up to 5% and backcattering ratios of around 1.10 to 1.15, this layer is driven by an enhancement of background aerosol (Vernier et al., 2011). While overall the amount of non-sulfate aerosol or anthropogenic sulfur dioxide sources are difficult to assess (Kremser et al., 2016), the ATAL might be responsible for about 15% of

the aerosol content in the Northern Hemisphere. This is comparable with all volcanic eruptions from 2000 to 2015 (Yu et al., 2017). Chemical in-situ measurements revealed that the ATAL mainly consists of secondary substances containing nitrate, ammonium, sulfate and organics and is therefore more diverse in its chemical composition than the Junge layer (Appel et al., 2022).

The Southeast Asian monsoon has very strong deep convection, where updrafts of 4 $^\mathrm{m}/_\mathrm{s}$ are common (Wang, 2004), into altitudes of 400 K (Randel et al., 2010; Bian et al., 2020) which allow both gas-phase aerosol precursors and aerosol particles from surface sources to reach the stratosphere (Vogel et al., 2019; Brunamonti et al., 2018; Hanumanthu et al., 2020). Several studies confirm that the horizontal transport out of the Asian monsoon anticyclone has an impact on the lower extra-tropical stratosphere of the Northern Hemisphere (Vogel et al., 2016; Yu et al., 2017, e.g.). Beside of volcanic activities, anthropogenic

emissions make a large contribution to the total concentration of aerosol with sulfuric components. As Notholt et al. (2005) show, the sulfur emission of Europe, North America and the former Soviet Union are constantly decreasing, while the emission drastically increases in Southeast Asia, where also the very effective uplift of pollution, gas-phase aerosol precursors and aerosol particles in the stratosphere by the Asian summer monsoon takes place. Some observational evidence of an increasing of stratospheric aerosols globally was found by Vernier et al. (2011) who used satellite data by CALIPSO observations that

revealed the ATAL near 16 km in June to September is associated with the Asian monsoon anticyclone.

Using the CLaMS model (Chemical Lagrangian Model of the Stratosphere) and aircraft measurements over Europe and the northern Atlantic, horizontal pathways out of the Asian monsoon anticyclone were found exporting young polluted air masses eastward into the extra-tropical northern lower stratosphere within several weeks (e.g. Vogel et al., 2016; Wetzel et al., 2021; Lauther et al., 2022). In particular source regions of anthropogenic short-lived species such as $CH_2Cl_2$ emission are mainly





located in Asia, therefore CH$_2$Cl$_2$ is a very good marker for transport out of the Asian monsoon anticyclone into the northern extratropical stratosphere (Lauther et al., 2022). The good linkage between CLaMS surface origin tracers and measured pollution tracers such as CH$_2$Cl$_2$ and PAN (Wetzel et al., 2021; Lauther et al., 2022) show that CLaMS is very well suited for analysing aerosol transport out of ATAL.

Furthermore, aerosol originating from extended boreal forest fires especially in Russia and North America have been seen in the Arctic summer stratosphere (Ohneiser et al., 2021; Cheremisin et al., 2022). This observation is becoming more frequent in recent years (Zielinski et al., 2020). According to satellite-based studies of global extinction results wild fires can only be found in the lowermost part of the stratosphere with a relatively short lifetime (Thomason and Knepp, 2023). Therefore, the annual cycle with its natural variability and the origin of the Arctic stratospheric aerosol layer is important.


Currently stratospheric aerosol represents an interesting research topic as already several studies analyse a potential impact of stratospheric aerosol injections on geoengineering (e.g. Irvine and Keith, 2020; Cheng et al., 2022), to counteract the global rise of temperature. As Sun et al. (2020) point out, the impact of an injection of sulfuric aerosols into the stratosphere will change globally the incoming radiation but also the precipitation. Additionally, the effect will be different depending on the

location of injection as well as the amount.

However, clearly all potential impacts on the climate system need to be understood prior to such an execution of costly and risky environmental endeavour. To minimise impacts of precipitation suppression some authors favour aerosol injection into the Arctic summertime stratosphere (Lee et al., 2021). It is therefore important to understand the existing contribution of the un-modified stratospheric aerosol content in the Arctic, which may also originate from the Southeast Asian monsoon. It is

therefore important to understand the seasonal variability of the Arctic stratospheric background aerosol impacted by the Asian summer monsoon.

Therefore we use model simulations to identify the contribution of the Southeast Asian monsoon to air masses detected by Lidar measurements in the Arctic. For this study we focus on the year 2021, since it was a year with a statistically average

amount of forest fires on the northern hemisphere and we have a qualitatively good Lidar data set for every month. We present an overview of basic stratospheric aerosol properties above our Arctic site in Section 3.3 which shows a weak but clear annual cycle. Further we show the output of a Lagrangian chemical transport model in Section 3.5 which shows the occurrences of air which originated in the Asian Summer monsoon region over the Arctic. The discussion of the results will be presented in Section 4 and the conclusion will follow in Section 5.



## 2 Measurement Site, Instrument and Model

### 2.1 Arctic Measurement Site: Ny-Ålesund

The observation site in Ny-Ålesund is located in the European Arctic (78.923°N, 11.928°E) on the northwest coast of the archipelago of Svalbard along the shore of Kongsfjord, which is orientated in a SE to NW direction on the west coast of Svalbard (Figure 1).

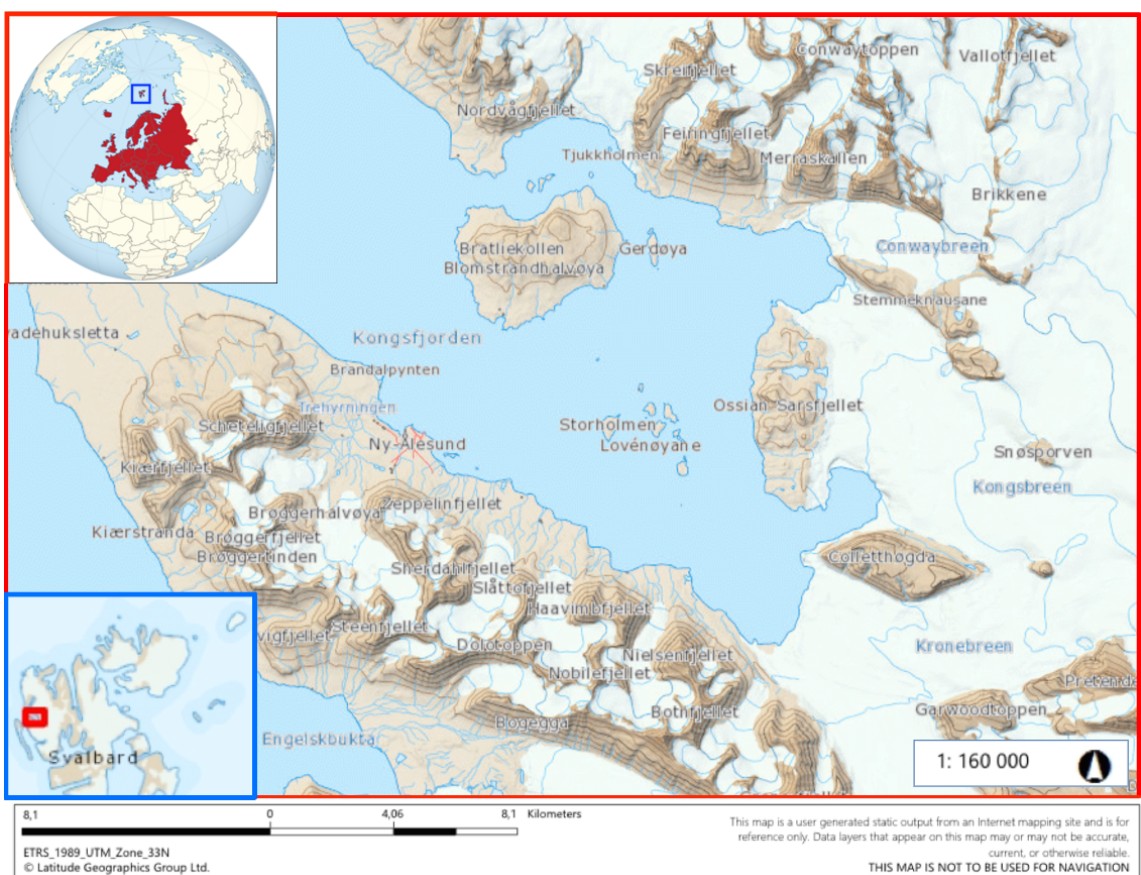

**Figure 1.** Map of the vicinity of Ny-Ålesund, Kongsfjord and its location in Europe. Sources: https://geokart.npolar.no/Html5Viewer/ index.html?viewer=Svalbardkartet (accessed on February 27, 2024), courtesy of Norsk Polarinstitutt and https://de.wikipedia.org/wiki/Datei: Europe_on_the_globe_(red).svg (accessed on February 27, 2024)



## 2.2 KARL – Koldeway Aerosol Raman Lidar

### 2.2.1 Description of KARL

The Lidar measurements were performed at the German-French AWIPEV research base in Ny-Ålesund, Svalbard by the "Koldewey Aerosol Raman Lidar" (KARL). This system consists of a Spectra 290/50 Nd:YAG laser, which emits a laser beam at 355 nm, 532 nm and 1064 nm, with 50 Hz and 200 mJ per pulse and color vertically into the atmosphere. The backscattered photons are collected by a 70 cm telescope with a field of view of about 2 mrad in the elastic colours as well as in the Raman-scattered signals at $N_2$ at 387 nm and 607 nm. Data recording is done via Hamamatsu photomultipliers and Licel transient recorders with 7.5 m resolution. Full overlap is reached at about 700 m altitude. A more detailed description of the instrument is given by Hoffmann (2011). The calculations of the aerosol backscatter coefficient $\beta^{aer}(\lambda) \left[ (\mathrm{m} \cdot \mathrm{sr})^{-1} \right]$ and the aerosol extinction coefficient $\alpha^{aer}(\lambda) \left[ \mathrm{m}^{-1} \right]$ were calculated following the principle of Ansmann et al. (1992). For better readability and simplicity we use $\beta(\lambda) := \beta^{aer}(\lambda)$ and $\alpha(\lambda) := \alpha^{aer}(\lambda)$ in the following, if not stated otherwise. The Lidar ratios, LR [sr], for the other wavelengths were chosen $LR_{355} = 50$ sr for the Troposphere and $LR_{355} = 70$ sr in the stratosphere, for $\lambda = 532$ nm $LR_{532} = 36$ sr and $LR_{532} = 42$ sr in the troposphere and stratosphere, respectively, if not stated otherwise. Since we do not expect rapid changes in the physical and chemical composition of aerosols in the lower stratosphere and to reduce the uncertainty due to noise, especially during polar day, we chose a temporal and spacial resolution of 60 min and 150 m for the elastic profiles at 355 nm and 532 nm. Both signals can always be evaluated up to almost 30 km height.

### 2.2.2 Parameters and Fundamental Equations

It is well known from Klett (1981, 1985) that the solution of the elastic Lidar equation is:

$$P(z) = Const \cdot \frac{\beta^{tot}(z)}{z^2} \cdot \exp \left( -2 \int_{z_0}^{z} \alpha^{tot}(\hat{z}) d\hat{z} \right) \tag{1}$$

where $P(z)$ is the signal power at a given height $z$, $Const$ a constant determined by the instrument itself and its setup, $\beta^{tot}(z) = \beta^{aer} + \beta^{Ray}$ the total backscatter signal and $\alpha^{tot}(z) = \alpha^{aer} + \alpha^{Ray} + \alpha^{abs}$ the total extinction, depends on the choice of a Lidar ratio and on the choice of a boundary condition. The extensions $aer$, $Ray$ and $abs$ are the contributions of aerosols and Rayleigh scattering as well as the absorption by trace gases. $\beta^{aer}$ needs to be prescribed at a certain altitude. Klett (1981, 1985) pointed out that the solution is stable and less critical dependent on the assumed boundary condition, when the integration is done "backwards": from large distances back to the Lidar.

We set the boundary condition, $\epsilon$, in the altitude interval from 27 km to 30 km for March to November. In winter times, Polar Stratospheric Clouds (PSC) might occur in this altitude (Tritscher et al., 2021), so we lowered the fitting range to 24 km to 27 km. In this altitude range we assume, the average total backscatter coefficient in that interval is $\epsilon_{532} = 1.10 \cdot \beta^{Ray}$ and



$\epsilon_{355} = 1.05 \cdot \beta^{Ray}$. The signal-to-noise ratio, $SNR$, is calculated following Equation 2:

$$SNR(z) = \frac{P(z)}{P_{err}(z)} = \frac{P(z)}{\sqrt{P(z) + P_{bgrd}}} \tag{2}$$

Where $P_{err}$ is the estimated noise profile and $P_{bgrd}$ the averaged signal in an altitude range, which is dominated by electronic noise and background light in an altitude range of 52 km to 60 km. The Lidar signal $P(z)$ is given in the units of "counts" like in the photo-counting channels. This altitude range has a sufficiently small median signal-to-noise ratio. Therefore it is valid to assume there is just noise in the signal. The SNR is shown in Table 1 at an altitude of 20 km for the different wavelengths and directions of polarisation.

**Table 1.** Mean of the signal-to-noise ratio for all emitted wavelengths and directions of polarisation at 20 km altitude

| $\lambda$ [nm] | $355_{\parallel}$ | $355_{\perp}$ | $532_{\parallel}$ | $532_{\perp}$ | 1064 |
|---|---|---|---|---|---|
| **SNR** | 52.7 | 7.5 | 15.6 | 10.6 | 4.06 |

With these high values of signal-to-noise ratio we conclude, that the data set is sufficiently good for a further evaluation of observations of physical features in the lower stratosphere. Since the depolarisation is very low (see Section 3.3.2), the signal is much weaker than for the parallel polarised light and therefore the SNR is generally smaller. Due to the small SNR at 20 km the signal of $\lambda = 1064$ nm is neglected for this study.

From the Lidar data we derive the following quantities: The Lidar ratio $LR$, aerosol depolarisation, $\delta^{aer}(\lambda)$ and the backscatter color ratio, $CR$, were calculated following Equation 3:

$$LR(\lambda) = \frac{\alpha^{aer}(\lambda)}{\beta^{aer}(\lambda)} \qquad \delta^{aer}(\lambda) = \frac{\beta^{aer}_{\perp}(\lambda)}{\beta^{aer}_{\parallel}(\lambda)} \qquad CR(\lambda_1, \lambda_2) = \frac{\beta^{aer}(\lambda_1)}{\beta^{aer}(\lambda_2)} \tag{3}$$

Here, $\beta^{aer}_{\perp}$ and $\beta^{aer}_{\parallel}$ are the backscatter coefficients with respect to the perpendicular and parallel laser polarisation, respectively, which are measured separately in the elastic 355 nm and 532 nm channels. For $\beta^{aer}$ and $\delta^{aer}$ only 532 nm and 355 nm signals were used. The color ratio is defined with $\lambda_1 < \lambda_2$.

In contrast to the backscatter coefficient, $\beta$, the backscatter ratio, R, is dimensionless and does not decrease with increasing height. It is defined as in Equation 4:

$$R(\lambda) = \frac{\beta^{tot}(\lambda)}{\beta^{Ray}(\lambda)} = 1 + \frac{\beta^{aer}(\lambda)}{\beta^{Ray}(\lambda)} \tag{4}$$

An ideal, aerosol-free atmosphere would have $R = 1$ and increases with stronger pollution. The parameter R is therefore a good indication for aerosol existence.



With the ERA5 reanalysis data by Hoffmann and Spang (2021) the mean height of the tropopause was found in an average

height of 9137 m at the study site of Ny-Ålesund during the Lidar measurement time and will be further discussed in Section 3.2. To investigate the properties of stratospheric aerosol, we focus on the altitude range between 10 km to 15 km; most of the troposphere is neglected and not shown in the plots.

### 2.2.3 Data Quality Management of Lidar Data

For this study, we only used automatically flagged data, which fulfilled the following criteria. Each wavelength and profile is tested and eventually removed individually:

1. Elimination of negative or too many ($> 20$) too low values ($R \leq 1.003$)

2. Elimination of artificial too high values $\left(\beta^{aer}(\lambda) \geq 5 \cdot 10^{-8} (\text{m sr})^{-1}\right)$, when the error and parameter variables are equally large, more than $^2/_3$ of the data points are not-a-number or, if in more than $^2/_3$ of the cases values of $R \geq 2.0$ is

160 reached

3. Clouds were detected in the troposphere with the criteria of to adjacent data points reach the limit of $\Delta\beta = 1 \cdot 10^{-4} (\text{m sr})^{-1}$

A qualitatively good profile has to fulfil all of these criteria at the same time. If one is not satisfied, the profile is eliminated out of the further processed data set. These criteria were chosen by manually looking at the data of different times of the year and atmospheric conditions with the aim of finding a good agreement between eliminating as many disturbed profiles as

necessary and keeping the good ones.

### 2.3 RS41 Radiosondes

At least once a day a radiosonde of the type RS41 by Vaisala is launched from AWIPEV at 11 UT. The data is processed according to the standards by the Global Climate Observing System (GCOS) Reference Upper-Air Network (GRUAN). A more detailed description of the radiosondes is given by Maturilli and Kayser (2017). With the quality checked and homogenised

data set the Rayleigh atmosphere for the correction of Lidar profiles is calculated as well as the tropopause height.

### 2.4 CLaMS – Chemical Lagrangian Model of the Stratosphere

To analyse the origin of air masses over Ny-Ålesund in 2021, we conducted 3-dimensional global simulations with the Chemical Lagrangian Model of the Stratosphere (CLaMS; McKenna et al., 2002b, a; Pommrich et al., 2014, and references therein) as well as CLaMS back-trajectory calculations starting along the Lidar measurements. The global model simulations are driven by

175 horizontal winds from ERA5 reanalysis (Hersbach et al., 2020), provided by the European Centre for Medium-Range Weather Forecasts (ECMWF).

We use ERA5 data in lower resolution (ERA5 $1° \times 1°$) (similar to Ploeger et al., 2021; Konopka et al., 2022; Clemens et al.,



2023). Here, ERA5 data are truncated to a $1° \times 1°$ horizontal grid and a 6-hourly time resolution, whereby the vertical reso-
lution is the same as in the original ERA5 reanalysis. ERA5 $1° \times 1°$ data are a computing-time-saving alternative to the full
resolution ERA5 data and are used for 3-dimensional global CLaMS simulations. Vertical velocities are calculated following
the diabatic approach by Ploeger et al. (2010). CLaMS uses a hybrid vertical coordinate ($\zeta$). At pressure levels lower than
300 hPa, $\zeta$ can be interpreted as potential temperature ($\theta$). Towards higher pressure levels, $\zeta$ transforms from an isentropic to a
pressure-based orography-following hybrid coordinate. Global 3-dimensional CLaMS simulations are based on 3-dimensional
forward trajectories and a parametrization of small-scale mixing depending on the shear in the large-scale wind flow (e.g.
Pommrich et al. (2014)).

A 3-dimensional CLaMS simulation starting from 1st December 2020 until the end of 2021 to cover the entire time period
of the year 2021 was performed including artificial surface origin tracers (Vogel et al., 2015, 2016, 2019). The surface origin
tracers used in this study cover the entire Earth's surface and are associated to 32 defined regions in the model boundary layer
that is about 2 km to 3 km above surface considering orography. They are released at the model boundary layer every 24 hours
and are further transported (advected and mixed) to the free atmosphere. Here in this study was found, that regions in South
Asia and in the tropical Pacific region (see Figure 2) have the major impact on air masses over Ny-Ålesund. The impact of all
other regions of the world are minor and are summarised as 'residual' and 'residual ocean'. Air masses from South Asia are
during summer associated with the Asian monsoon circulation and are here used as a marker for Asian monsoon air. Besides
of continental regions in South Asia in addition parts of the Indian Ocean and the Bay of Bengal are included into the surface
origin tracer for South Asia.

Starting with the onset of the Indian monsoon at the beginning of June, air masses from the Indian Ocean are transported
long-range to the North of the Indian subcontinent where they are uplifted into the upper troposphere and lower stratosphere
(e.g. Fadnavis et al., 2013; Lau et al., 2018; Nomura et al., 2021; Vogel et al., 2023). These air masses can uptake polluted
air while passing over the Indo-Gangetic Plain where anthropogenic emission are higher compared to other regions in India
caused by the dense concentration of industries and the high population (e.g. Fadnavis et al., 2016). Therefore, the tracer for
the Indian Ocean and the Bay of Bengal are included into the marker for the Asian monsoon air.





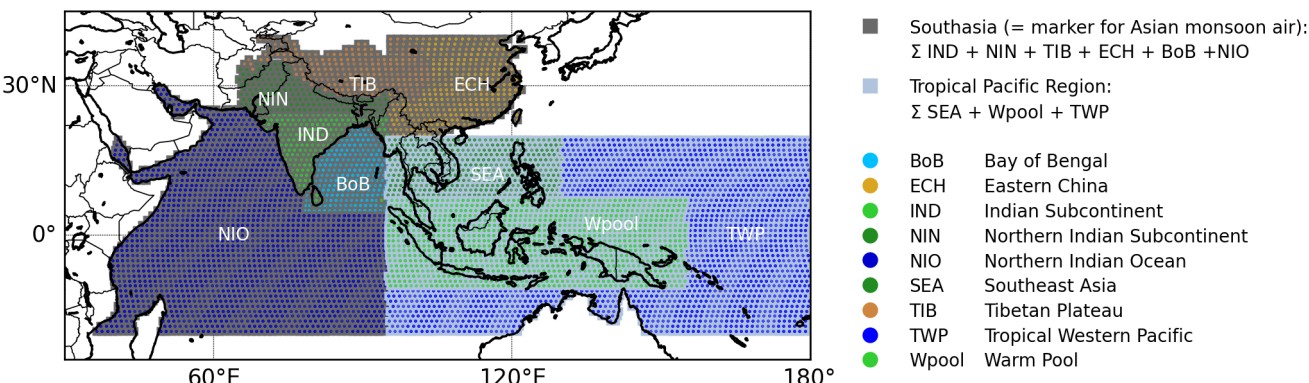

**Figure 2.** Geographical map showing the location of CLaMS's surface origin tracers in South Asia and the tropical Pacific region. Regions that mostly impacting the Lidar measurements in Ny-Ålesund are combined into two tracers referred to as 'South Asia' and 'tropical Pacific region'. The names of the surface origin tracers and their corresponding abbreviations are listed right beside the map

## 2.5 Moderate Resolution Imaging Spectroradiometer – MODIS

Moderate Resolution Imaging Spectroradiometer (MODIS) is a NASA satellite-based radiometer for Earth observations in 36 different spectral bands reaching from 0.4 $\mu$m to 14.4 $\mu$m. It has, depending on the selection of bands, a spatial resolution of $0.25° \times 0.25°$ and a temporal resolution of about two days. Wild fires are detected by the 4 $\mu$m and 11 $\mu$m bands by observing temperature anomalies relative to the background and absolute. In this study only the parameter Fire Radiative Power (FRP) from Aqua and Terra were used to characterise wild fires. More information can be found at MODIS.

## 3 Results


### 3.1 Wild Fires in North America and Russia

Wild fires occur regularly in the Northern Hemisphere and are a big source for aerosol, which can be lifted by several processes into the stratosphere (Ansmann et al., 2018; Ohneiser et al., 2021). Since air masses coming from Southeast Asia pass Russia and potentially even Canada the contribution of wild fires has to be considered, if the observed increased backscatter is influ-
enced by these or is actually a direct signal from the monsoon region. The area "Russia" was defined within the rectangular defined by the geographical coordinates 77°N 31°E and 48°N 180°E. "Canada" is defined by the corners at 71°N 170°W and 48° N 52°W. Active wild fires are combined to monthly overviews. Figure 3 gives an overview about the median and mean Fire Radiative Power (FRP) measured by MODIS and averaged over all grid cells in the defined regions "Canada" and "Russia" from 2001 to 2022.





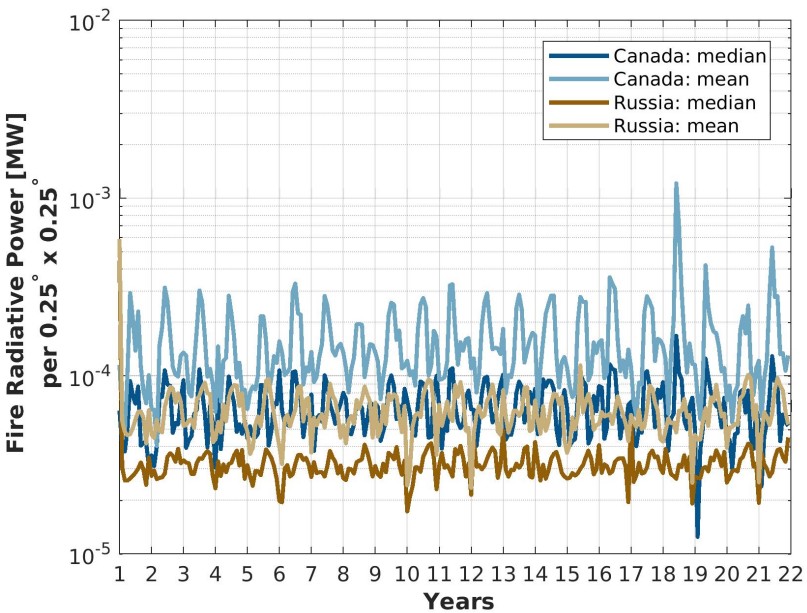

**Figure 3.** Overview of monthly averaged over the entire domain of either "Canada" or "Russia" fire radiative power (FRP) from 2001 – 2022

The wild fire season is in general more pronounced in Canada than in Russia throughout a year after averaging over the entire domain, since the variability from year-to-year is larger. It can be seen that the observed fire radiative power per grid cell does not increase during the observed 21 years, neither for Russia nor for Canada. The year 2021, which is selected for this study, shows a slightly increased higher FRP than other years in "Canada", but not in "Russia". Generally, strong forest fires produce clearly visible aerosol layers in Lidar observations, which are visible at least for several weeks (e.g. Zielinski

et al., 2020; Ohneiser et al., 2021; Cheremisin et al., 2022). Since the aerosol layers from wildfires can reach altitudes of 20 km and more (Ohneiser et al., 2023), they would also influence our Lidar observations. We have not found such layers in our data set of 2021. Nevertheless the possibility must be considered that aerosol from biomass burning events lead to a higher stratospheric background in the Arctic in summer and autumn. We therefore conclude, that wildfires did not play a major role in the stratospheric aerosol budget in 2021.

**3.2   Tropopause Calculations**

The tropopause is a transport barrier between the troposphere and the stratosphere. We used the daily launched radiosonde from AWIPEV as well as ERA5-reanalysis data with a hourly global resolution of $0.3° \times 0.3°$ to determine the tropopause height for every Lidar profile. A full description of the two data sets are given by Maturilli and Kayser (2017) and Hoffmann and Spang (2021), respectively. As the definition of the tropopause the lapse rate definition, $\frac{dT}{dz} \leq 2\frac{\mathrm{K}}{\mathrm{km}}$, of the World Meteorology

Organisation (WMO) (Organization, 1957) is used for both data sets.





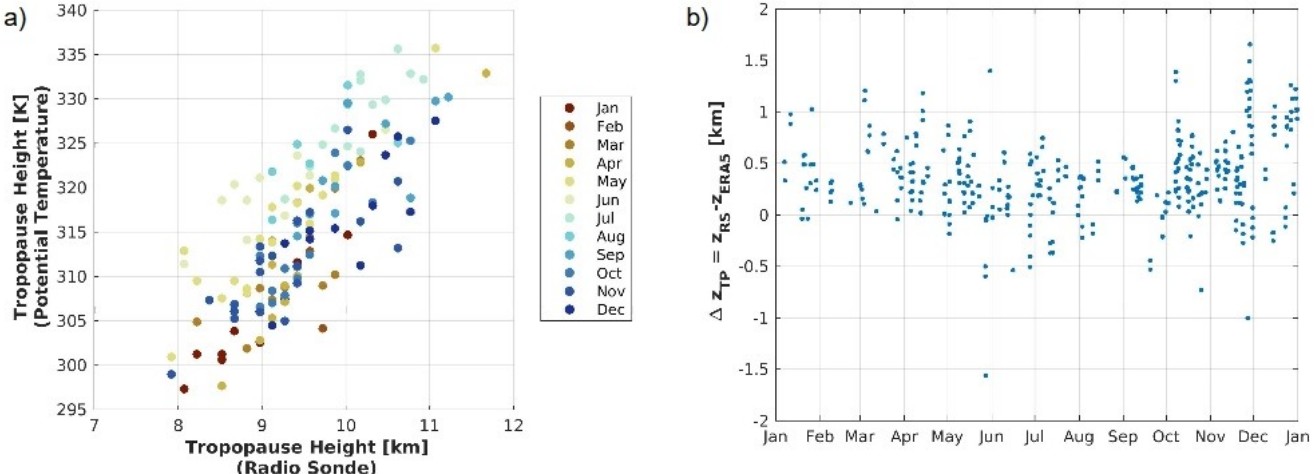

**Figure 4.** Annual cycle of Tropopause (TP) height measured by radiosondes in relation to the potential height of the tropopause derived from radiosondes (Figure 4a). The deviation between radiosonde tropopause height and ERA5 data is given in [km] (Figure 4b)

The tropopause height is shown in Figure 4. Since the measurements of radiosonde and Lidar are based on the geometric height, we transformed the altitude into potential temperature for better comparison with CLaMS results that is based on isentropic coordinates in the stratosphere. Figure 4a shows the geometric height in [km] and compares it with the corresponding potential temperature at this point, $\theta$:

$$\theta = T \left( \frac{p_0}{p} \right)^{R/c_p} \tag{5}$$

Where $T$ and $p$ are the current temperature $[K]$ and pressure $[hPa]$ of the air parcel respectively, $p_0 = 1013$ hPa the ground reference pressure, $R$ the gas constant of air and $c_p$ the specific heat capacity. We set $R/c_p = 0.286$.

Every data point in Figure 4a corresponds to a measurement day. For comparison we also calculated the geometric tropopause with ERA5 data and compared it with radiosonde data throughout 2021 (Figure 4b). Since there is only one radiosonde launch a day, the Lidar measurement time and the launch does not always match, but the deviation between both measurements is maximum 12 h. On the other hand the data by Hoffmann and Spang (2021) has an hourly resolution corresponding to the chosen Lidar resolution. The median tropopause height in the radiosonde data is at 314.5 K (9011 m) and on average 329 m above the results of the ERA5-reanalysis model.

It can be seen, that in general the tropopause has a higher potential temperature than in winter, but there is also a variability within a month. We conclude nevertheless that the tropopause is always at $\theta < 340$ K and aerosol above it is always in the stratosphere with increasing geometric height during summer months. Generally, the annual cycle of the tropopause observed by radiosondes is well represented in the ERA5 data.



### 3.3 Observations by KARL

In the following study we just concentrate on the year 2021, since we have 481 h for 532 nm and 474 h for 355 nm of qualitatively good Lidar measurement time and is available throughout the entire year as well as every month. Due to additional campaign activity, especially in November and December, and due to high cloud cover fraction especially in summer (Graßl et al., 2022), the absolute number of measurement time varies a lot between months and seasons. February has the smallest amount of good data with 3 h, while in November 103 h of measurement time is available. An overview of the monthly mea-

surement time [h] is given in Table 2. Since the quality checks are performed for each emitted wavelength independently, a deviation in the available amount of measurement time might occur.

**Table 2.** Overview over the monthly available measurement time [h] for 355 nm and 532 nm after quality check

|         | Jan | Feb | Mar | Apr | May | Jun | Jul | Aug | Sep | Oct | Nov | Dec | Total |
|---------|-----|-----|-----|-----|-----|-----|-----|-----|-----|-----|-----|-----|-------|
| 355 nm  | 22  | 3   | 22  | 32  | 35  | 43  | 44  | 16  | 44  | 87  | 102 | 24  | 474   |
| 532 nm  | 21  | 3   | 22  | 34  | 42  | 44  | 44  | 17  | 44  | 85  | 102 | 23  | 481   |

Figure 5 shows the monthly medians of two aerosol backscatter coefficients $\beta_{532}$ and $\beta_{355}$ for the selected four heights of the lower stratosphere with a thickness of 20 K, which corresponds to about nine vertical measurement points with the given

resolution.





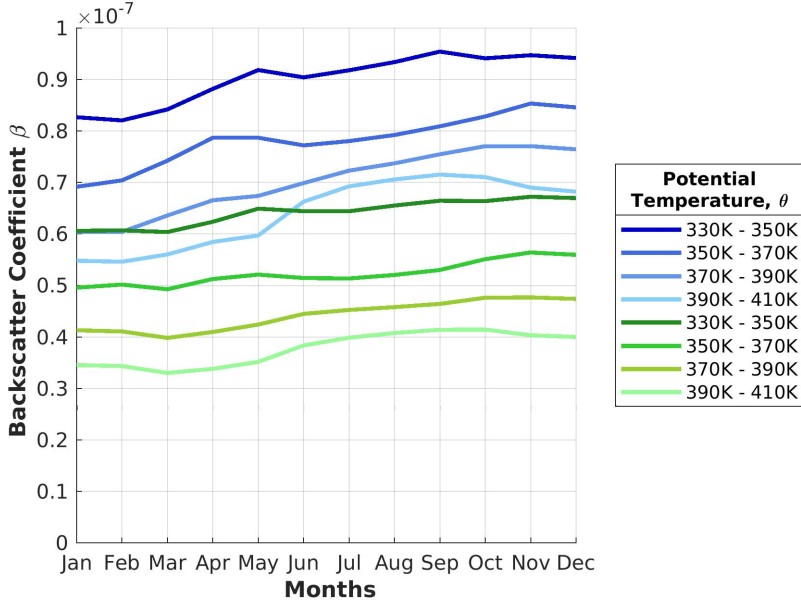

**Figure 5.** Monthly median backscatter values for $\beta_{532}$ (green colors) and $\beta_{355}$ (blue colors) for the lower stratosphere (330 K to 410 K potential temperature)

Generally, a weak annual cycle with minimal values of backscatter in spring and highest values in autumn can be seen. However, the curves are quite smooth, especially for 532 nm. At 400 K the amplitude of the annual variability is around 27% and 25% for 355 nm and 532 nm, respectively.

For all four selected layers $\beta_{355}$ is always larger than $\beta_{532}$. Further, the backscatter decreases noticeably with increasing potential temperature. While the annual cycle of the backscatter in the lower two altitude ranges seems to be more variable and "disturbed", with a secondary maximum already in April or May.Between 370 K and 410 K, an increase of both backscatter coefficients is found from June to October 2021. As the backscatter and the amplitude of the annual cycle is larger in the UV we conclude that these changes refer to smaller particles, since the scattering efficiency strongly depends on the particle radius

(Mie, 1908). The estimation of the effective radius of stratospheric aerosol by Mie calculations will follow in Section 3.4. Exemplary days will be later discussed and linked to CLaMS simulations in Section 3.5.

### 3.3.1 Color Ratio

The color ratio, $CR$, is a crucial parameter to determine the size of aerosols and is defined according to Equation 3. An overview of the analysed year is shown in Figure 6. As a ratio of two small quantities, $CR$ critically depends on noise and numerical

assumptions in the Lidar evaluation (Foken, 2021). Hence, some noise is noticeable in form of rapid changing coloured dots in Figure 6.





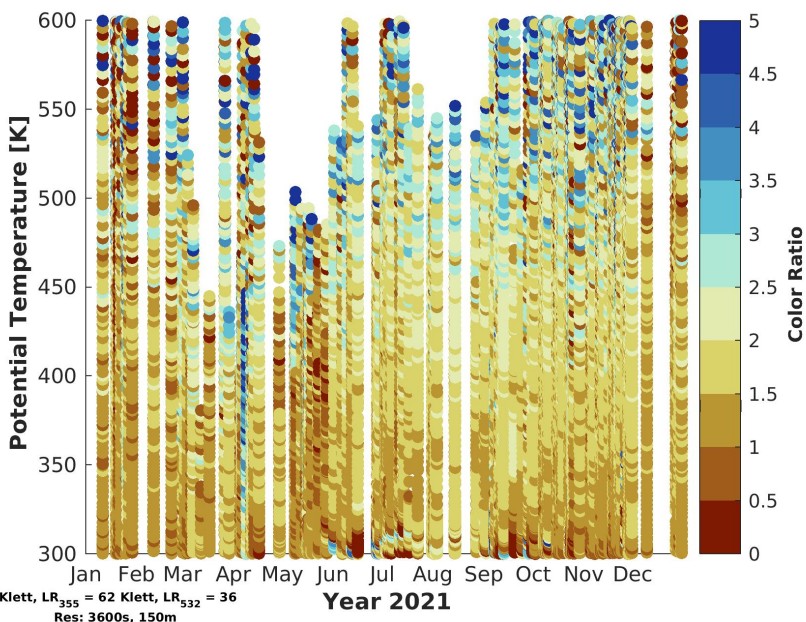

**Figure 6.** Overview of the color ratio, $CR$, throughout 2021

Typically, CR-values around 2 have been found. Remarkable is the fact that the second half of the year becomes more homogeneously with slightly larger values than the first half. From June to October a more homogeneous distribution of a $CR \sim 2$ is detected between 320 K and 440 K potential temperature. Since $CR$ is a function of particle radius, the particle size also decreases towards the end of the year in the discussed altitude range of 330 K to 410 K, but also with increasing height as it was also seen by Junge et al. (1961a). According to Murphy et al. (2021) the generally larger particles (lower $CR$) in the first half of the year may be an indication of reduced tropospheric advection in the Arctic lower stratosphere in this time of year.

The relationship between $\beta_{532}$ and $CR$ is depicted for the previously defined four layers of the lower stratosphere in Figure 7. With this dependency a better understanding of the particle size and concentration can be made.





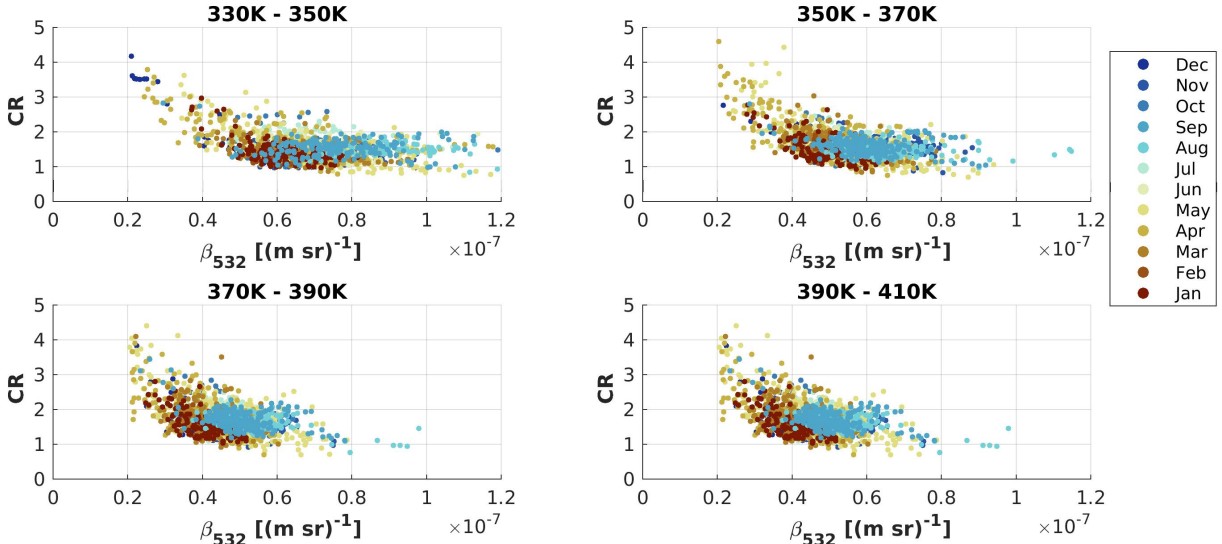

**Figure 7.** Dependency of $CR$ and $\beta_{532}$ for four selected height intervals of the lower stratosphere

While the spread in $\beta_{532}$ decreases with height, $CR$ remains in this same range of about $CR \in [1, 4]$. In altitudes of 330 K – 350 K a small influence of the troposphere might be seen in summer months, since the tropopause can reach these altitudes exceptionally. In the range of $CR \in [1, 2]$ all altitude layers have a backscatter coefficient larger at the end of the year than in the beginning. Under pristine conditions, $\beta_{532} \leq 5 \times 10^{-8} (\mathrm{m\,sr})^{-1}$ we see always a clear dependence between $CR$ and $\beta$: The smaller $\beta$ (the clearer the atmosphere) becomes the larger CR becomes, because small particles show only a small scattering efficiency. Hence, below $\beta_{532} = 5 \times 10^{-8} (\mathrm{m\,sr})^{-1}$ the stratospheric $\beta$ is mainly driven by the size of aerosols. This effect levels to above the limit of $\beta_{532} = 5 \times 10^{-8} (\mathrm{m\,sr})^{-1}$. Larger than a certain limit the stratospheric aerosol does not grow apparently. This is in agreement to the annual cycle already shown in Figure 5. Since the color ratio does not change much in the four discussed intervals it also means the aerosol size is very similar, but only the concentration becomes lower the higher up the layer is located.

### 3.3.2 Aerosol Depolarisation

Another parameter to determine physical properties of aerosol by remote sensing is using the so-called Aerosol depolarisation, $\delta^{aer}(\lambda)$. It is defined according to Equation 3. The laser of KARL is linearly polarised and is always the reference direction of the scattered light. With Mie theory it can be shown, that spherical particles do not change the polarisation of the laser light, while non-spherical ones do. Due to surface tension liquid droplets are in general spherical (Foken, 2021).

In general stratospheric aerosols have the property of $\beta_{\perp}^{aer} \ll \beta_{\parallel}^{aer}$. Therefore, it is expected that $\delta^{aer}$ becomes quite noisy for a standard aerosol measurement in higher altitudes. Due to generally low values of the depolarisation we only show and





discuss $\delta_{532}$ in this study.

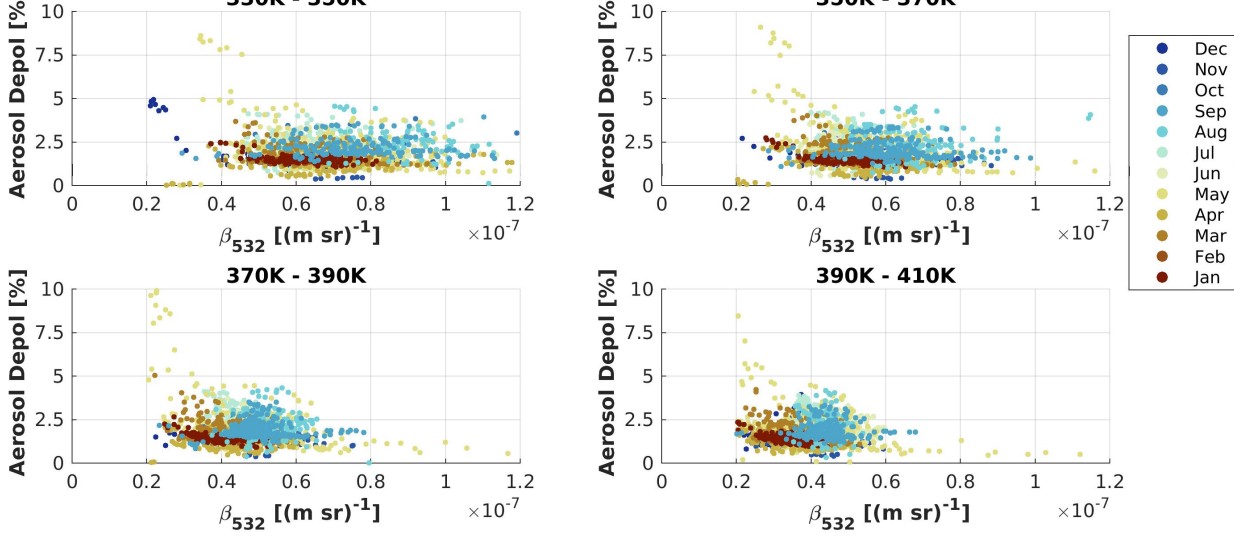

**Figure 8.** Dependency of $\beta_{532}$ and $\delta_{532}$ for the four selected altitudes in the lower stratosphere

The overview of the aerosol depolarisation, $\delta_{532}$, is given in Figure 8 for the different layers of the lower stratosphere color coded for every month. Due to case studies in Section 3.5 September is highlighted already here. The depolarisation is low ($< 10\%$) for all corresponding values of $\beta_{532}$. All four height intervals a common depolarisation is between $\delta_{532} \geq 1\%$ and $\delta_{532} \leq 4\%$. Additionally May is the most divers month with single cases of $\delta_{532} \rightarrow 10\%$, while January shows the smallest

depolarisation. In general, summer has a higher depolarisation than winter months.

We define a "weak" depolarisation with $\delta_{532} \geq 2\%$ (see Table 3) a "medium" depolarisation with $\delta_{532} \geq 5\%$ (see Table 4). A "strong" depolarisation with $\delta_{532} \geq 10\%$, but was not found in the lower stratosphere of 2021.

**Table 3.** Relative occurrence of at least a "weak" polarisation ($\delta_{532} \geq 2\%$) in different altitude layers of the lower stratosphere. All numbers are given in [%]

|  | Jan | Feb | Mar | Apr | May | Jun | Jul | Aug | Sep | Oct | Nov | Dec |
|---|---|---|---|---|---|---|---|---|---|---|---|---|
| 330 – 350 K | 3.81 | 0 | 21.43 | 7.29 | 13.94 | 13.96 | 25.76 | 49.33 | 17.50 | 3.96 | 0.39 | 4.89 |
| 350 – 370 K | 3.81 | 0 | 22.92 | 5.83 | 14.83 | 14.02 | 31.82 | 50.56 | 14.17 | 1.92 | 0 | 4.80 |
| 370 – 390 K | 2.91 | 0 | 18.67 | 6.85 | 17.18 | 15.28 | 27.53 | 34.67 | 12.36 | 1.68 | 0 | 5.78 |
| 390 – 410 K | 3.89 | 0 | 24.29 | 3.91 | 18.17 | 16.90 | 24.01 | 32.50 | 9.58 | 0.94 | 0 | 18.40 |



A big variability between the months can be observed in Table 3 for depolarisation of $\delta_{532} \geq 2\%$. While the depolarisation is in general lower in the winter months, it reaches its maximum in August and has a clear annual cycle. This indicates that the stratospheric aerosol observed above Ny-Ålesund is not constant throughout the year but changes its chemical composition and / or its shape (Foken, 2021). Furthermore, a more frequent depolarisation is found in lower altitudes than in higher ones. Our Lidar observations fit very well to the physical properties of aerosols in the Asian Tropopause Aerosol Layer (ATAL), which consist of organics, nitrate, sulfate and ammonium, especially in combination to Figure 8, where it can be seen, that the aerosol is also slightly non-spherical (Appel et al., 2022). The altitude, in which this aerosol layer in the Arctic was found also matches very well to the altitude of the ATAL.

**Table 4.** Relative occurrence of at least a "medium" polarisation ($\delta_{532} \geq 5\%$) in the lower stratosphere from 330 K – 410 K. The numbers are given in [%]

|             | Jan | Feb | Mar  | Apr | May  | Jun | Jul | Aug | Sep  | Oct | Nov | Dec  |
|-------------|-----|-----|------|-----|------|-----|-----|-----|------|-----|-----|------|
| 330 – 410 K | 0   | 0   | 0.10 | 0   | 2.08 | 0   | 0   | 0   | 0.15 | 0   | 0   | 1.16 |

Table 4 shows on the other hand the relative occurrence of $\delta_{532} \geq 5\%$ for the entire layer of 330 K – 410 K. Only in a few months a "medium" depolarisation were observed. Even though the depolarisation has a annual cycle all values are very small compared to lower latitudes (Foken, 2021). No young extreme pollution events, like wild fire aerosol, dust intrusion or volcanic ashes with typical values of $\delta_{532} \geq 16\%$ (Foken, 2021) were found in the Lidar measurements over Ny-Ålesund in 2021. Therefore we conclude that our data set indeed mainly represents stratospheric background conditions even with the contribution of the ATAL. Therefore we conclude that our data set represents stratospheric conditions free from local events, however there is evidence of a seasonal impact of ATAL particles to the lower stratosphere during summer.

Figure 9 shows for two months with the lowest (February) and highest median depolarisation (August) in comparison for the lower stratosphere and further investigate the difference between the relative occurrence of depolarisation.





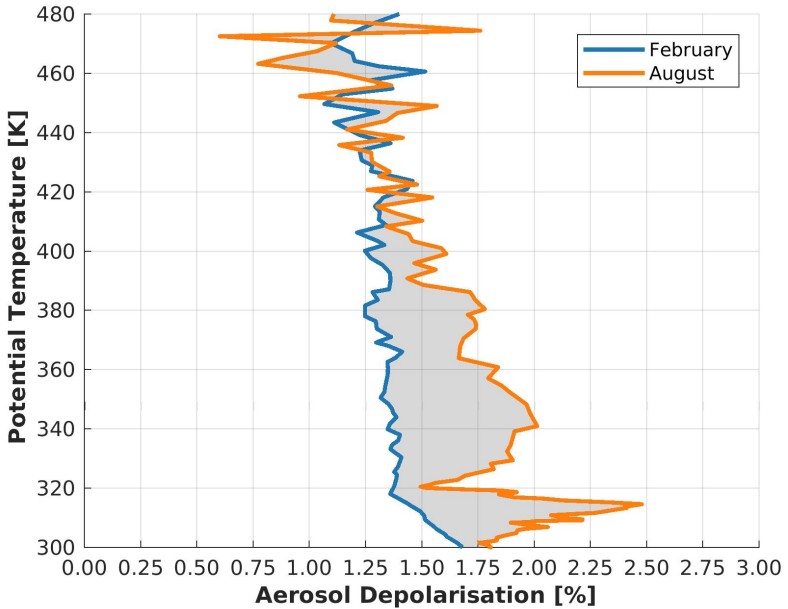

**Figure 9.** Mean aerosol depolarisation, $\delta_{532}$, for the two exemplary months February and August. The grey shaded area shows the difference between both months

While February has a more or less constant depolarisation at $\delta_{532} = 1.35\%$ in the atmosphere between 320 K and 480 K, August on the other hand has in general higher depolarisation values, especially in the lower parts of the atmosphere (until about 420 K). The grey shaded area highlights the difference between both months. Especially in the troposphere and lower stratosphere a significant deviation can be found and shows the impact of the ATAL. Additionally, the summer month also

has a maximum around 315 K, where it reaches $\delta_{532} = 2.5\%$, followed by a second, local maximum at around 340 K with $\delta_{532} = 2.0\%$. From that point on it decreases to about $\delta_{532} = 1.25\%$ and converges to the same value as in February. Due to solar radiation in August, the data becomes noisy quicker than during polar night. At an altitude of 400 K the depolarisation in August is still significantly higher than in February. The median tropopause heights according to radiosonde data is in February at 307 K height, while it is at 324 K in August. This plot also shows that even if no distinct forest fire layer has been seen in

our Lidar data during summer, the slightly less spherical shape of the particles indicates a non-sulfuric component that is an evidence for an impact of ATAL particles on the Arctic background aerosol up to ~450 K in August 2021, even though aged forest fire aerosol may still contribute to the stratospheric aerosol background, which were even found in altitudes of up to 27 km (Ohneiser et al., 2023).



### 3.3.3 Seasonal variability of stratospheric Arctic background Aerosol

Our measurements indicate evidence for the impact of ATAL particles on the Arctic stratosphere during summer, therefore we now present the annual cycle of $\beta_{355}$ and $\beta_{532}$ of the background aerosol in Figures 10 and 11, respectively, for two different heights of 360 K and 380 K. The red marks, $-$ and $+$, represent the monthly median and mean respectively. The blue box indicates the 25th and 75th percentiles, while the black lines stand for the 9th and 91th percentiles. Data from $\pm 10$ K around the chosen level are taken into account for the following study.

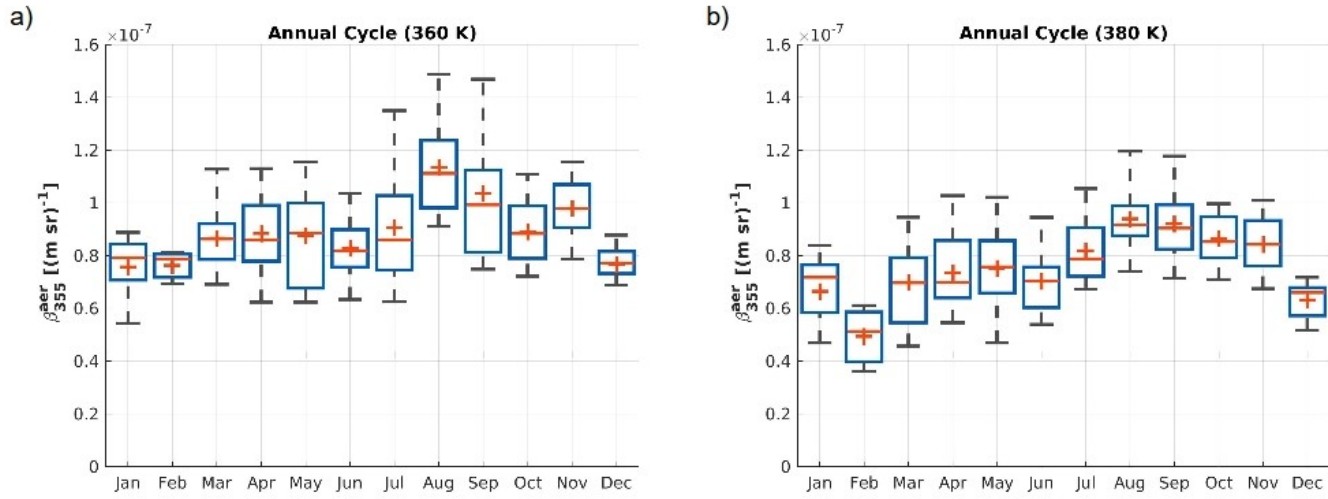

**Figure 10.** Annual cycle of stratospheric aerosol backscatter coefficient $\beta_{355}$ in 360 K and 380 K height. The red symbols, $-$ and $+$, show median and average, respectively. The blue box symbolise the 25th and 75th percentiles, respectively. The black lines show accordingly the 9th and 91th percentiles

The annual cycle of $\beta_{355}$ in 360 K altitude (Figure 10a) shows a small seasonal dependency: While the first half of the year is characterised by lower values of the backscatter coefficient, the mean values of August, September and November are $> 10^{-7} (\mathrm{m\ sr})^{-1}$. While winter (DJF) is the clearest season, May, July and September are the months with the broadest distribution. Overall the median and mean values are for all months very similar, indicating that $\beta_{355}$ follows almost a Gaussian distribution.

Likewise there is a similar annual cycle in the data at the 380 K level (Figure 10b) with larger values from June to November. Also here winter (DJF) is the clearest season. While the absolute values of $\beta_{355}$ are consistently lower than at 360 K height, the relative spreading of $\beta_{355}$ is also smaller. Typical values are for 360 K altitude are $\beta_{355} \in [7, 11] \times 10^{-8} (\mathrm{m\ sr})^{-1}$, they are smaller for the 380 K-level, $\beta_{355} \in [6, 10] \times 10^{-8} (\mathrm{m\ sr})^{-1}$. In summary, a clear annual cycle in the stratospheric backscatter coefficient has been found.

 

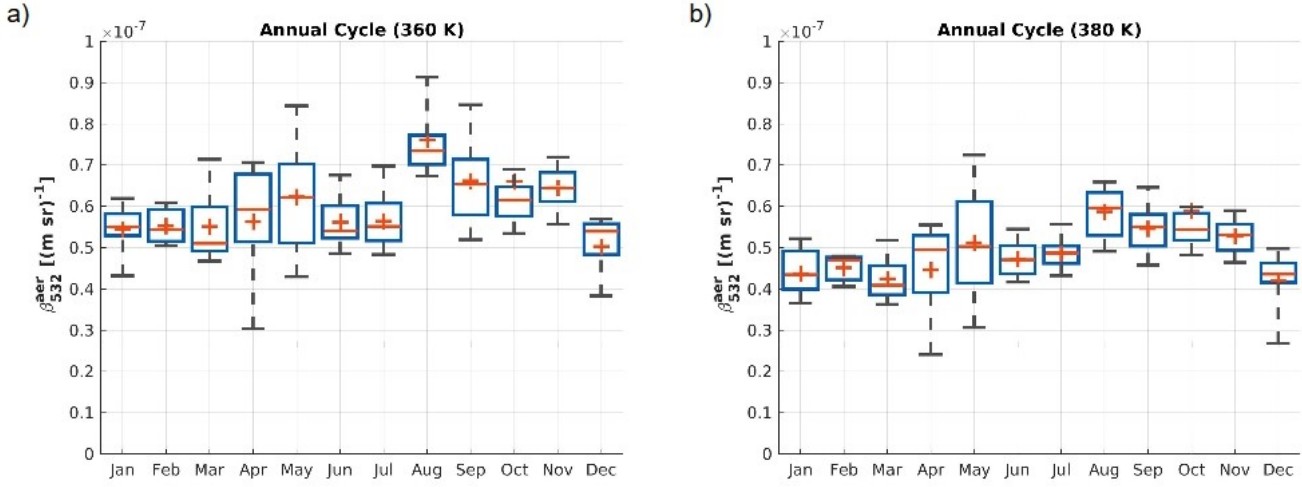

**Figure 11.** Annual cycle of stratospheric aerosol backscatter coefficient $\beta_{532}$ in 360 K and 18 km height. The red symbols, $-$ and $+$, show median and average, respectively. The blue box symbolise the 25th and 75th percentiles, respectively. The black lines show accordingly the 9th and 91th percentiles

While the values of $\beta_{532}$ are by about a factor of 2 smaller than for $\beta_{355}$ a similar pattern is shown in Figure 11a compared to $\beta_{355}$ with some differences: The variability in the first half of the year is more pronounced and the increased backscatter coefficient peaks in August. On the other hand $\beta_{532}$ is less homogeneously distributed than $\beta_{355}$ throughout the entire year. The 380 K-level (Figure 11b) shows the same trend throughout the year with a higher month-to-month variability in April and May.
Median and mean values of $\beta_{532}$ spread more than for $\beta_{355}$, indicating a more asymmetrical distribution and more variability within a month. Typical values are for 360 K altitude $\beta_{532} \in [5,7] \times 10^{-8}(\text{m sr})^{-1}$ and $\beta_{532} \in [1,6] \times 10^{-8}(\text{m sr})^{-1}$ for the 380 K-level.

### 3.4 Mie Calculus for Particle Size Estimations

Absorption and scattering of a single spherical particle surrounded by a non-absorbing medium takes is described by Mie scattering (Mie, 1908). Particles can have a complex refraction index, $m$, in general:

$$m = n + i\,k \tag{6}$$

The scattering efficiency factor is a parameter, which is defined as the ratio of the scattering cross section, $\sigma_{eff}$ (Equation 7), to the geometrical cross section. Therefore, it is a good measure to determine the effective radius for a mono-modal log-normal
aerosol size distribution:



$$\sigma_{eff} = \pi \int\limits_0^\infty Q_{sca}^{FF}(r)\, r^2 f_1(r)\, dr \tag{7}$$

The scattering efficiency $Q_{sca}^{FF}(r)$ for the far field, $FF$, and $f_1$ the mono-modal log-normal aerosol size distribution following the calculations according to Bohren et al. (2023).

We expect small or already aged aerosols in the stratosphere (Turco et al., 1982; Kremser et al., 2016) and are therefore able to estimate the effective radius of these particles by the Library for Radiative Transfer – libRadtran (Mayer and Kylling, 2005; Emde et al., 2016). As Kremser et al. (2016) pointed out, sulfur chemistry is a very common and frequent process in the stratosphere. Since sulfate containing particles are not the only aerosol type in the ATAL, we also chose biomass burning particles as the second example as a highly absorbing particle type.

We roughly estimate an effective radius for the observed aerosol, which can have a lifetime of up to several weeks (Oppenheimer et al., 1998). Since the refractive index is wavelength-dependent two different indices were chosen for sulfate: $m_{532}^{Sulfate} = 1.44 + 10^{-4}i$ (Russell and Hamill, 1984; Yue et al., 1994) and $m_{355}^{Sulfate} = 1.41 + 0i$. The imaginary part of Equation 6 can be neglected for $m_{355}^{Sulfate}$, since sulfate does not absorb in UV (Beyer and Ebeling, 1998; Washenfelder et al., 2013). As a comparison we also calculated the effective radii for a strongly absorbing aerosol type, like from wild fires and chose a

refractive index of biomass burning aerosols (BB) of $m^{BB} = 1.52 + 10^{-2}i$ for 355 nm and 532 nm.

With the above mentioned assumptions we used the two refractive indices for 355 nm and 532 nm, a log-normal distribution of particles with a standard deviation of $\sigma = 1.5$ up to a maximum size of 600 nm and a bin with of the distribution of 10 nm.



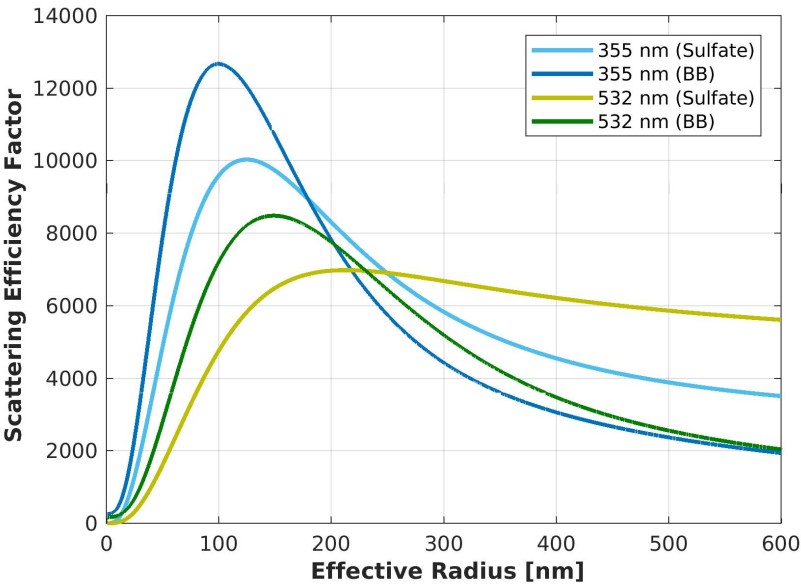

**Figure 12.** Scattering efficiency factor, $Q_{sca}^{FF}$, in arbitrary units for the emitted 355 nm and 532 nm laser beam for a log-normal distribution of purely sulfate particles and strongly absorbing BB aerosols

As Figure 12 shows, the larger the real part of the refractive index the quicker the scattering efficiency drops for larger effective radii. Simultaneously the maximum of scattering efficiency moves towards larger $r_{eff}$ for larger real parts and wave-
lengths. On the other hand, there is not a big difference in scattering efficiency for large particles especially for biomass burning aerosol observed in the wavelengths 355 nm and 532 nm. Even though the efficiency is much lower for smaller particles in 532 nm. An overview of the different effective radii correlated to typical color ratios (see Section 3.3.1) are given in Table 5.

**Table 5.** Calculated effective radii, $r_{eff}[nm]$ corresponding to typical Color Ratios with $m^{BB} = 1.52 + 10^{-2}i$, $m_{532}^{Sulfate} = 1.44 + 10^{-4}i$ and $m^{Sulfate}355 = 1.41 + 0i$

| $r_{eff}$ | $CR$ | | | |
|---------|------|------|------|------|
| $[nm]$ | 2.00 | 2.25 | 2.50 | 2.75 |
| Sulfate | 97 | 82 | 70 | 59 |
| BB | 84 | 72 | 61 | 52 |

It is expected to find a small effective radius for stratospheric aerosols distributions, since the particles form in chemical reactions in Aiken mode on salt solutions or salt embryos. The reaction takes also place at stratospheric temperatures of
$T \leq -55°C$ (Friend et al., 1973). With these small effective radii corresponding to an height dependent value of $CR$ (see Figure 6) we conclude, that in general the effective radius of aerosol in the lower stratosphere is relatively small (Murphy



et al., 2021) and the particles are sorted according to their size by for example sedimentation processes with an accumulation of largest particles close to the tropopause.

## 3.5 Origin of air masses inferred from CLaMS simulations

Backscatter coefficients, color ratio and depolarisation of the aerosol particles detected by KARL show evidence that the ATAL has an impact on the Arctic lower stratosphere. To confirm this assumption we link KARL measurements to Lagrangian transport simulations. To identify the source regions of the aerosols detected by KARL, global 3-dimensional CLaMS simulations including surface origin tracers were analysed and discussed for two exemplary days in September 2021 (13th and 28th). The identification of possible source regions of the aerosol particles found above Ny-Ålesund (Section 3.5.1) are investigated by

global 3-dimensional CLaMS simulations including surface origin tracers (Section 3.5.2) as well as CLaMS back-trajectories (Section 3.5.3) .

### 3.5.1 Seasonal variability of air mass origins over Ny-Ålesund

The contributions of young air masses from South Asia and the tropical Pacific over the course of the year 2021 to the air over Ny-Ålesund is inferred for the heights $(360 \pm 5)$ K (Figure 13a) and $(380 \pm 5)$ K (Figure 13b), respectively. Mean values are

calculated for the surface origin tracers for each day.

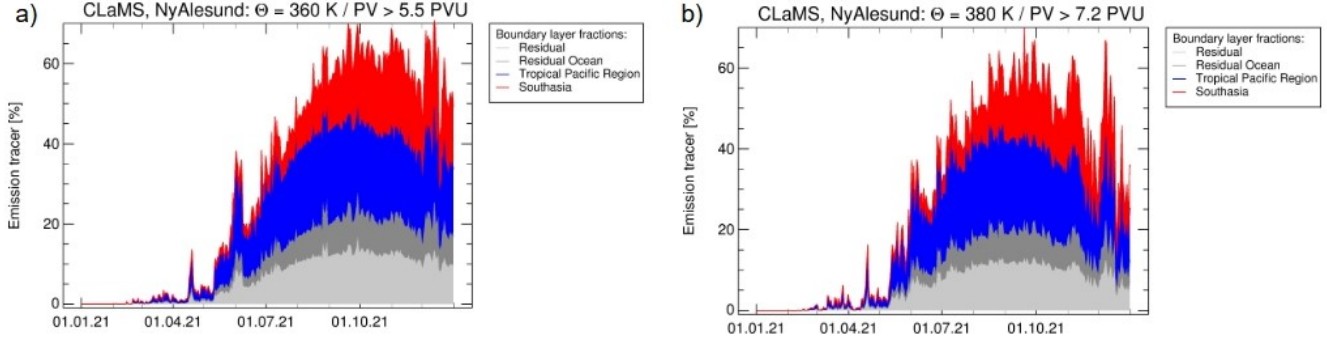

**Figure 13.** Contribution of different surface emission tracers from South Asia, Tropical Western Pacific, Residual and Residual Ocean to the lower stratosphere above Ny-Ålesund at 360 K (Figure 13a) and 380 K (Figure 13b) potential temperature for the year 2021. The 5.5 PVU and 7.2 PVU surface takes the climatological isentropic transport barrier at 360 K and 380 K, respectively, into account (Kunz et al., 2015)

Artificial tracers released in the model simulations since 1st December 2020 in Figure 13 do not reach Ny-Ålesund at levels of potential temperature of 360 K and 380 K before May 2021. During summer and autumn, the highest fractions of tropospheric air over Ny-Ålesund are from South Asia and the tropical Pacific region. Especially in September and October almost 20% of the air is from South Asia. These model results support the evidence that the Arctic lower stratosphere is

impacted by aerosol particles from the ATAL (Yu et al., 2017; Bian et al., 2020). All other regions combined are of minor importance. With few exceptions the "Tropical Pacific" contributes constantly more, sudden changes in its impact are rare.



The accumulation of young air masses in the lower stratosphere over Ny-Ålesund is lower 100% during the year 2021. The remaining fraction consists of aged air, that is older than 1st December 2020 when the CLaMS simulation was initialised.

### 3.5.2 Three-dimensional CLaMS simulation

Figure 14 gives for two exemplary days in September (13th and 28th) the distribution of the surface origin tracer for South Asia (a marker for air from the Asian monsoon region) over northern Europe at 360 K and 380 K potential temperature. A high fraction from South Asia is found over the Arabian Peninsula marking the western outflow of the Asian monsoon anticyclone (e.g. Vogel et al., 2016). Air masses contributing to the ATAL can be transported eastwards along the subtropical jet and enter the lower stratosphere by quasi-horizontal transport. As a result, thin filaments with enhanced signatures of air mass tracers

origin from South Asia were found over the northern Atlantic Ocean and Europe (e.g. Vogel et al., 2016; Wetzel et al., 2021; Lauther et al., 2022). On 28 September a filament located over Ny-Ålesund with contribution from South Asia up to 40% is simulated. This filament was not found on 13 September.

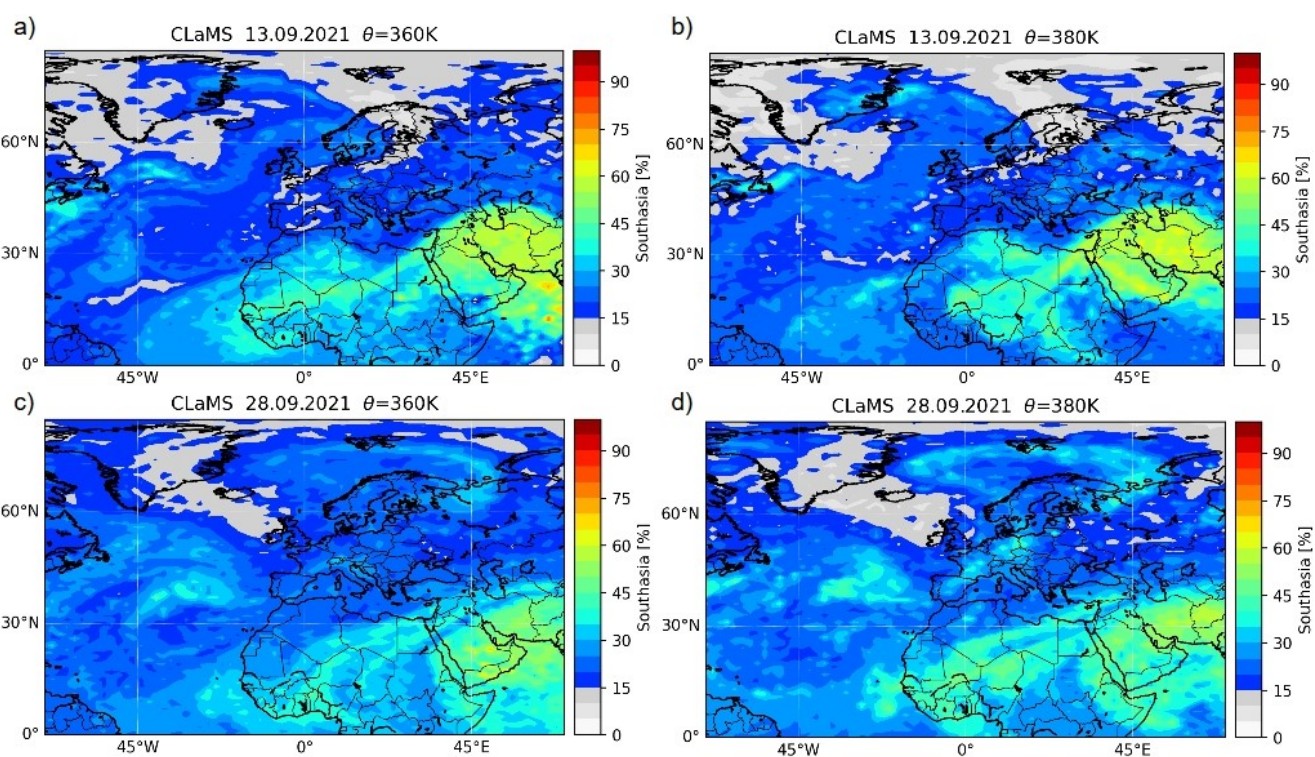

**Figure 14.** Horizontal distribution of the fraction of air originating in South Asia (details of the region can be found in Figure 2) at 360 K (left column) and 380 K (right column) potential temperature on 13 (top), and 28 (bottom) September 2021



Figure 14 shows in the stratosphere from Southeast Asia for the two different heights of 360 K and 380 K. The contribution
is quite low on 13 September in 360 K in 380 K altitude (Figures 14a and 14b). As it can be seen in both figures, the contri-
bution is constant over entire Europe, but increases with increasing height and changes on time scales of less than a day. As it
can be seen in both altitudes the impact is already very low over Svalbard ($\approx 10\%$). On 28 September a filament was seen faint
in 360 K altitude (Figure 14c) but very pronounced in 380 K (Figure 14d). The presented values are strong for Ny-Ålesund up
to 20% are comparable small compared to other sites at lower latitudes.

When comparing Figures 13 and 14 with each other the contributions of the two main sources of tracers can be identified.
The region 'Tropical Pacific' is responsible more for the background amount, which can be observed in Figure 14. Contrary
to this, 'Asia' has a smaller contribution but shows also a very large day-to-day variability. These spikes can be nicely seen as
filaments in the three dimensional simulations of tracers of Figure 14.

The following preliminary conclusion can be made: There is a large day-to-day variability in the distribution of young air
masses from Asia contributing the Arctic lower stratosphere. This variability strongly depends on the dynamics of the Asian
monsoon anticyclone itself and the transport of air masses from Asia to the extra-tropical lower stratosphere.

### 3.5.3 Case studies using back-trajectory calculations

Backward trajectories are well suited to analyse the detailed transport pathway of an air parcel and therefore complement the
three-dimensional CLaMS simulations (e.g. Vogel et al., 2014; Wetzel et al., 2021; Lauther et al., 2022). Within this study,
we calculated an ensemble of backward trajectories for the two exemplary days (13 and 28 September 2021). Within $\pm 2°$ in
latitudinal and longitudinal direction around Ny-Ålesund and in a height range of 360 K to 380 K, 891 trajectories have been
calculated in total for each day. Markers in Figure 15 show the location, where the trajectory first reaches the boundary layer.
The color indicates the transport time.





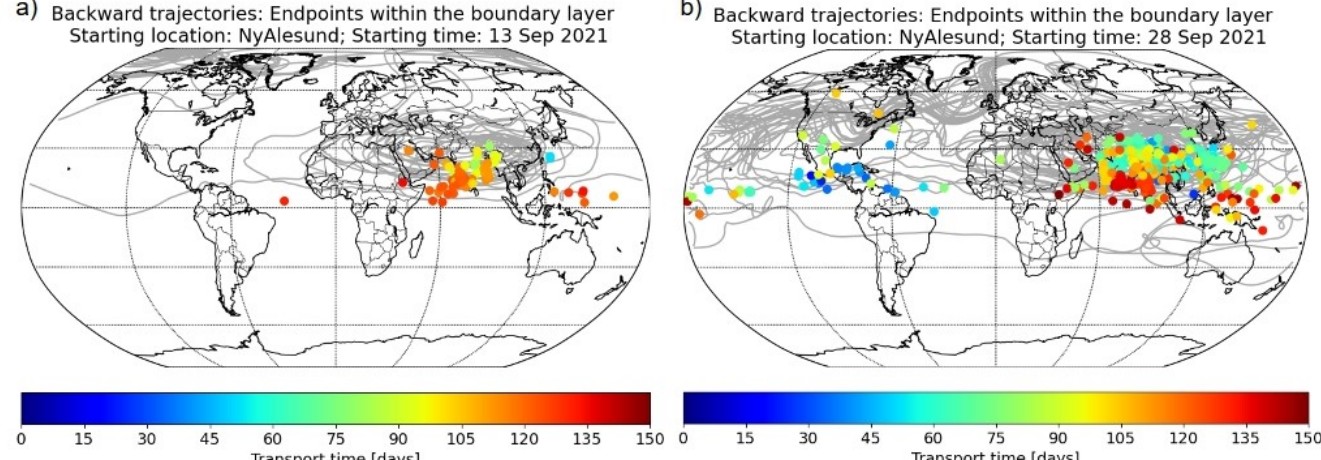

**Figure 15.** Location of trajectory end points where they reach the model boundary layer (BL). Trajectories are initialised on 13 and 28 September within 360 K and 380 K in altitude and ±2° horizontally around Ny-Ålesund. The end points are colour-coded by the transport time between reaching the Boundary Layer the first time and the Lidar measurement at Ny-Ålesund

As it can be seen in Figure 15a on 13 September in total 86 trajectories (9.6%) reached the boundary layer in Southeast Asia after travelling for about three to four months, 7 other trajectories (0.8%) are outside of the monsoon region. 798 trajectories (89.6%) did not touch the boundary layer at all within the model run time.

The situation is a bit different for 28 September 2021 (Figure 15b), where the fraction of contacts outside of the monsoon region is larger (7.6%), but also the fraction of end points in Southeast Asia (37.9%). On this day only 54.4% of the 891

trajectories did not reach the model boundary layer within the computational time. It took the air parcel on average 87 days to reach Ny-Ålesund on 28 September. As it can be seen on both days, back-trajectories also end in different parts of the world, like the equatorial pacific ocean but also central America. This result agrees very well with the forward trajectories of aerosol tracers of Section 3.5.2. The average transportation time from the South Asian region to Ny-Ålesund is about one month longer on 13 September (111 days) than on the 28 September (87 days).

The presented numbers are also given in Table 6 for a better overview with the absolute and relative numbers of back-trajectories touching the boundary layer inside and outside of the monsoon region in South Asia.



**Table 6.** Contact points of all calculated back-trajectories, which are presented in Figure 15 with their relative and absolute frequencies as well as the mean propagation time between the South Asian monsoon region and Ny-Ålesund

|  | 13 Sep 2021 | 28 Sep 2021 |
|---|---|---|
| **Boundary Layer** | 160 (18.0%) | 213 (23.9%) |
| **Asia** | 35 (3.9%) | 159 (17.8%) |
| **Residual** | 6 (0.7%) | 54 (6.1%) |
| **no contact** | 850 (95.4%) | 678 (76.1%) |
| **Sum** | 891 (100%) | 891 (100%) |
| **Mean transport time** | 111 days | 87 days |

Additionally one exemplary trajectory is given for each day in Figure 15. As it can be seen in both cases, the artificial tracer circulates from Ny-Ålesund first around the pole, before it moves to lower latitudes until it reaches South Asia, where it also circles, until it reaches the boundary layer. Since both trajectories have similar pathways, we conclude that the intake of aerosols into the lower stratosphere depends on the local conditions in South Asia and the effectivity for aerosols being lifted into the ATAL.

### 3.6 Detailed Analysis with KARL

In the following these two exemplary days will be discussed in detail. On one of them CLaMS observed a filament in different heights. The other day is an example with a comparable clear stratosphere. As a reference day a profile of February was additionally plotted to emphasise the difference between seasons and days. Here the backscatter ratio, defined according to Equation 4, is used as it does not decrease with increasing height and layers are easier to detect.





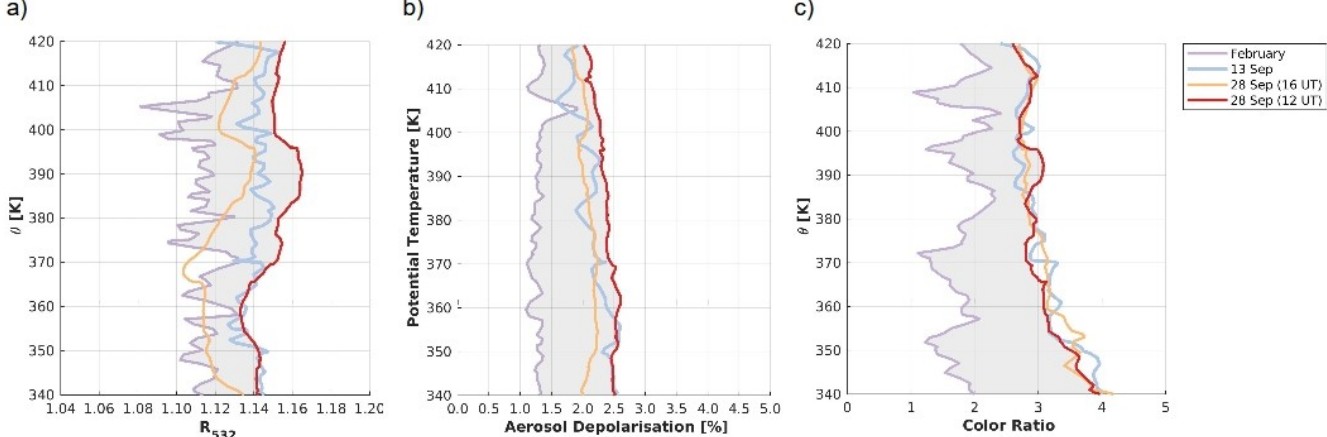

**Figure 16.** Profiles of different parameters, $R_{532}$ (Figure 16a), $\delta_{532}$ (Figure 16b) and $CR$ (Figure 16c), for the two exemplary days. 28 September is shown here twice, since the Lidar measurement simultaneously took place with the model output at 12 UT. As a comparison the Lidar results are also shown at 16 UT for 28 September. The shaded area highlights the difference between February and the filament on 28 September

Figure 16 shows for the two exemplary days in September with and without a detected filament by CLaMS, $R_{532}$ (Figure 16a), $\delta_{532}$ (Figure 16b) and $CR$ (Figure 16c). The shaded area highlights the difference between a typical profile of February, which is also not affected by the ATAL, and the detection of the filament on 28 September at 12 UT. Since the model
output of CLaMS is shown in Figure 14 for 12 UT and the size and movement of the filaments indicate only a short temporal increasing of the backscatter ratio, the measurements by KARL on 28 September are shown twice: for 12 UT and 16 UT. Due to the horizontal transports in the stratosphere the filament of this day has already moved out of the line-of-sight of the Lidar for the later time. According to the radiosonde data, the tropopause is well defined on the 28 September 2021 at 332.9 K. The reanalysis data of tropopause heights from the data repository Jülich Data (Hoffmann and Spang, 2021) reveals a tropopause
height of 329.1 K to 335.1 K over Ny-Ålesund for the two Lidar profiles. For this case the air masses of troposphere and stratosphere are well separated by each other and an exchange is not possible.

Generally, it can be seen, that all parameter, backscatter ratio, aerosol depolarisation and color ratio, are larger and less variable for all days in September than in February as it was already shown in Figures 10 and 11. This implies that the stratospheric
aerosol is already aged and well-mixed.

The backscatter ratio reveals a layer on the 28 September at 12 UT between 360 K and 400 K. This one is also seen in CLaMS as filaments (Figures 14c and 14d). KARL and CLaMS also agree that the layer is more pronounced at 380 K than at 360 K. According to KARL the layer reaches until 400 K altitude. Compared to 12 UT the layer at 16 UT has already a significantly decreased backscatter ratio, which is almost on the background level (defined as the mean backscatter ratio of the measure-
ments not affected by the filament). From Figure 16a it can be seen that the strongest impact of the filament occurred around



390 K. $R_{532}$ rises from about $R_{532} = 1.14$ to $R_{532} = 1.165$. This corresponds to an increase of aerosol backscatter by about 15%.

The depolarisation is in general quite small with values around 2.5% at 340 K, a maximum of all profiles at around 360 K. $\delta$ decreases afterwards with increasing height to values around 2% at 420 K. These small values indicate nearly spherical particles throughout the lower stratosphere. A slight increase in depolarisation throughout the entire lower stratosphere is four for 12 UT of 28 September, when the filament passes above the station.

The same result of small particles can also be found in the color ratio in Figure 16c. While $CR$ reaches values around 4 at 340 K, it decreases to $CR \approx 2.9$ in the altitude range of 370 K to 410 K. This means, aerosols are smallest and reach a constant size distribution. The filament of 28 September changes the overall color ratio only in the altitude range of 385 K to 395 K by $CR \approx 0.2$, meaning that the particle size is slightly smaller in times with a filament. February reveals a comparable low color ratio (larger particles), but also a lower backscatter ratio. We therefore conclude, aerosols are in general larger but less abundant in February than during September. The fluctuations are also larger, which indicates a less effective vertical mixing of the lower polar stratospheric air.

Observations by KARL and modelling studies by CLaMS reveal a good agreement between the results. Filaments, which were forecasted by CLaMS were found in KARL data. Physical properties of the stratospheric aerosol in the Arctic in September reveal very similar properties as in the ATAL. We can therefore conclude, the filaments above Ny-Ålesund originate from the ATAL. Stratospheric aerosol in other seasons, like February, has different physical properties and can clearly be distinguished from aerosols arriving in September.

## 4 Discussion

In MODIS data we did not find any hint, that 2021 was exceptionally in regards to the wild fire season on the northern hemisphere. The detailed studies of Canadian wild fires by Haarig et al. (2018) and Baars et al. (2019) reveal a aerosol depolarisation of up to $\delta_{532} \leq 20\%$ in the stratosphere. An explanation for the deviation of the aerosol depolarisation is the very low relative humidity in the stratosphere (Maturilli and Kayser, 2017). Since the aerosol remained there for already some days, it dried out and retrieved a spherical shape, with which a depolarisation of $\delta_{532} \approx 2\%$ is expected and was found in our data. A similar strong exemplary wild fire event, like it was analysed by Ansmann et al. (2018), was not found in our data.

In this work we showed a clear annual cycle of the stratospheric aerosol for the Arctic in the year 2021. In summer and autumn the aerosol backscatter is larger (by about 20%) than in winter and spring, especially in the UV. At ALOMAR (Arctic Lidar Observatory for Middle Atmosphere Research, northern Norway) a study about stratospheric aerosols was performed by Langenbach et al. (2019) for the years 2014 – 2017. The authors found a significantly increased backscatter ratio in the lower stratosphere (12 km to 18 km) in the months July and August for an emitted wavelength of $\lambda = 1064$ nm. Hence, the Arctic or sub-Arctic stratosphere is not pristine, especially aerosol contamination in summer may be typical. Compared to the strato-



spheric background aerosol at ALOMAR, we observe the aerosol layer in lower altitudes (in km) and the strongest appearance

in August and September. Deviations between these two measurement stations are probably due to the geographical location and the in general lower tropopause height in the Arctic than in lower latitudes.

After aerosol particles or their gas-phase precursors were uplifted into the lower stratosphere by the monsoon system, the air mass are confined within the Asia monsoon anticyclone from which eddies and filaments can be shed to the east (Weigel et al.,

2021a, b). These polluted air masses are further transported into the stratosphere of the Northern hemisphere by the breaking of Rossby waves (Waugh, 1996; Vogel et al., 2016), therefore it is expected to find filaments with an enhanced number of aerosol particles in higher latitudes. Aerosols contributing to the ATAL can circulate a few times around the Asian monsoon anticyclone before leaving Asia. In addition they can be transported along the subtropical jet a few time around the globe until they reach the Arctic. There it can be then further investigated by the Lidar KARL by assuming similar propagation velocities

as Jumelet et al. (2020). Afterwards the aerosol load decreases slowly by sedimentation and other removal processes happening in the stratosphere described in general by Junge et al. (1961a). From December to July the aerosol load is low and about constant. This is in good agreement with (e.g. Junge et al., 1961a; Kremser et al., 2016), who observed both an equilibrium state of new particle formation out of the gas phase and removal processes from the stratosphere into the troposphere. The physical properties of the filaments in the Arctic stratosphere reveal that the particles originate from the ATAL.


Since various types of aerosols are presented in the ATAL, we estimated the effective radii of two exemplary types: One with purely sulfate-containing particles, the other highly absorbing one from wildfires. Mie calculus revealed an effective radius of 52 nm to 97 nm as two more extreme examples of the aerosol classes. Since the depolarisation is comparably small, we also conclude that the predominant aerosol type has a spherical shape, which agrees to the in-situ measurements of Junge et al.

(1961a); Hofmann et al. (1975); Murphy et al. (2021) in lower latitudes or of aged aerosols emitted by wild fire events agree with the calculated effective radius of the in the Arctic observed aerosols (120 nm to 160 nm) and additionally enhances the assumption of a log-normal particle size distribution (Dahlkötter et al., 2014; Baars et al., 2019).

Taking now the height-dependent calculations of Junge et al. (1961a) into account sedimentation velocities of about $v_{min} = 3 \times 10^{-3} \frac{cm}{s}$ to $v_{max} = 2 \times 10^{-2} \frac{cm}{s}$ can be found. Rough estimations about the altitude loss of aerosols due to sedimentation

are presented in Table 7, with travelling times of $t_{min} = 60$ d to $t_{max} = 120$ d.

**Table 7.** Height loss of stratospheric aerosol due to sedimentation while travelling two to four months from the Southeast Asian monsoon region to Ny-Ålesund

|  | $t_{min}$ | $t_{max}$ |
|---|---|---|
| $v_{min}$ | 155 m | 311 m |
| $v_{max}$ | 1037 m | 2074 m |





Since the fall velocities are for these particles such small, it is probable that they are long enough in the stratosphere. Additionally, due to the slow sedimentation velocity as well as their small size (see Section 3.4), aerosol from the monsoon region can reach Ny-Ålesund.

According to CLaMS back-trajectory calculations for the two exemplary days (13 and 28 September 2021) air parcels from surface sources in Asia have transport times from about two to four months until they reach the lower stratosphere over Ny-Ålesund. Therefore, enhanced stratospheric backscatter coefficients measured from April to May 2021 are not caused by the Asian summer monsoon that usually starts at the beginning of June. Hence, the monsoon contributes to the aerosol load but is overlaid by other aerosol sources and the beginning of the increased backscatter coefficient is based on other contributions not

only by the monsoon. The Lidar data of 28 September in comparison with CLaMS agree very well that the monsoon signal can be clearly found in the Arctic as filaments by a noticeable increased backscatter ratio of about 15%.
Furthermore, the spreading between more and less polluted days is between August to October at the same level as in the other months (Figures 10 and 11). Therefore, we conclude that the increased backscatter coefficient in summer is caused by advection of aerosol from the Asian summer monsoon region and the advected particles from the ATAL. We clearly captured

one event where a filament of "monsoon air" increased the particle backscatter by about 15% at 380 K to 400 K (28 September, Figure 16a). The increased backscatter also happens about two months shown in Figures 10 and 11 after the monsoon in South Asia started. As it can be seen in Figure 15 it is possible for air parcels to travel within these two months from South Asia to Ny-Ålesund. Therefore we conclude, we can observe the signal of the South Asian monsoon in the Arctic stratosphere.

Aerosol model simulations by Yu et al. (2017) proposed that aerosol particles from the ATAL can spread throughout the entire extra-tropical northern lower stratosphere and contribute significantly ($\approx 15\%$) to the Northern hemisphere stratospheric column aerosol surface area on an annual basis. Eastward air mass transport of aerosol particles from the ATAL was already detected over Japan using Lidar measurements (Fujiwara et al., 2021). Khaykin et al. (2017) used long-term Lidar and satellite data over France and concluded that the ATAL is the main source for the increase of stratospheric aerosol in years without

major volcanic eruptions after the maximum of aerosol load in summer. Our findings, show that the ATAL during summer and autumn even has an impact on the Arctic lower stratosphere.
    Stratospheric aerosol pollution from the lower latitudes has already been observed over Antarctica as well. In the study of Jumelet et al. (2020) aerosols by Australian wild fires reached the stratosphere by strong convection in Pyro-Cumulonimbus (PyroCb). They were traced and measured by the satellite-based Lidar on CALIOP as well as by the Lidar operated at the

French Antarctic station Dumont d'Urville and cycled around the pole within about 6 weeks following the dominant wind fields. In slightly higher altitudes (400 K to 450 K altitude) smoke layers in the shape of filaments were detected over Antarctica, which passed by within days. The observations of Jumelet et al. (2020) perfectly agree with the observations by KARL and the model output by CLaMS.





## 5 Conclusions and Outlook

In this work an annual record of Lidar observations at AWIPEV in Ny-Ålesund has been presented which is free from obvious layers like polar stratospheric clouds (PSC), volcanic eruptions or forest fires. The Lidar measurements have been linked to Lagrangian transport simulations with the CLaMS model using artificial surface origin tracers as well as back-trajectory calculations. The main findings of this work can be summarised:

- The low stratosphere reveals an annual cycle with lower backscatter in winter until early spring and higher values in summer to autumn

- Between 380 K and 400 K altitude the backscatter coefficient is by 25% larger in summer compared to autumn for 532 nm. The annual cycle is even slightly more pronounced for 355 nm

- The increased backscatter coefficient in June to early August can be explained by the Asian monsoon due to the time, the air is advected. The observed increasing backscatter coefficient during summer and autumn correlates with the advection time and the beginning of the South Asian monsoon. Furthermore, the depolarisation shows a maximum during summer, even though it is generally low in our data set. Hence, probably biomass burning events contribute to stratospheric background aerosol even if no clear distinct layer can be seen by the Lidar

- Stratospheric aerosol generally shows a very low depolarisation, indicating almost spherical particles. Similar to the backscatter we found an annual cycle with slightly larger depolarisation in summer. The measured depolarisation as well as the back-scatter coefficient very well fit to the physical properties of the ATAL (Vernier et al., 2011).

- Lagrangian transport simulation with CLaMS show that air masses found between 360 K to 380 K air over Ny-Ålesund were transported from surface sources in Asia into the Arctic lower stratosphere. Case studies using back-trajectory calculations for two different days are presented. Our findings show, that the increased backscatter coefficient during summer can be linked to transport of aerosol particles from the Asian Tropopause Aerosol Layer (ATAL) into the Arctic lower stratosphere. Maximum backscatter coefficients were found during the peak season of the Asian monsoon in August. We demonstrate that the ATAL and thus the Asian monsoon has an impact on enhanced stratospheric aerosol backscatter coefficients found from June to October in the Arctic

- When assuming a mixed aerosol composition, which is typical of the ATAL, we chose optical properties of sulfate and biomass burning aerosol to calculate the aerosol effective radii using Mie calculation. In the first half of the year we obtained radii of 80 nm to 100 nm and 60 nm to 70 nm in the second half of the year

Our findings show that transport of aerosol particles from ATAL have an impact on the Arctic lower stratosphere. A regional radiative forcing of $-0.1$ W/m$^2$ caused by the ATAL was calculated from CALIOP measurements compensating for about one-third of the comparable radiative forcing associated with the global increase in CO$_2$. This regional radiative forcing due to the ATAL can be compared with the global aerosol forcing caused by moderate volcanic eruptions since 2000. argue that

further increasing industrial emissions in Asia will lead to a wider and thicker ATAL. An enhanced ATAL most likely yield increased transport of aerosol particles to the extra-tropical northern stratosphere that could enhance the climate impact of aerosol particles in this region in the future.

Since stratospheric aerosol has a very important impact on the climate, further investigations of the natural aerosol load are important. It would increase our knowledge of its contribution, its sources and ageing processes by adding more observation sites worldwide to the study. Stations at remote places and high altitudes would be preferred to minimise the tropospheric influence to the signal. Furthermore, in-situ measurements by plane and balloons at different locations and times within the lower stratosphere would also quantify the properties of aerosol better.


*Data availability.*    The data for the tropopause heights for 2021 can be found on the data repository Jülich Data including further information (Hoffmann and Spang, 2021, 2022). The radiosonde data is available at the PANGAEA repository (Maturilli, 2020). MODIS data about wildfires was taken from EarthData Repository. The Lidar evaluation software is written in MATLAB and can be obtained from the authors

*Author contributions.*    Lidar, MODIS, Radiosonde and ERA5 tropopause data have been evaluated by S.G. C.R. participated in Lidar evalu-
ation. The CLaMS simulations and analysis were performed by I.T. and B.V. All authors participated in the conceptualisation. The draft has mainly been written by S.G. All authors have read and agreed to the published version of the manuscript

*Competing interests.*    The authors declare no conflict of interest

*Acknowledgements.*    The Lidar KARL was serviced and operated by several Observatory Engineers and impres GmbH at AWIPEV, Ny-Ålesund. We also thank Roland Neuber and Marion Maturilli as scientific coordinators of AWIPEV, who supported the Lidar project over all
the years



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
