# Peer review of "Does the Asian Summer Monsoon Play a Role in the Stratospheric Aerosol Budget of the Arctic?"

_EGUsphere, 2024_

## Referee Comment (RC1)

The manuscript by Graβl et al, with Title "Does the Asian Summer Monsoon Play a Role in the Stratospheric Aerosol Budget of the Arctic?" analyses the contribution of Asian Tropopause Aerosol Layer during the Asian summer monsoon to the lower polar stratosphere about Ny-Ålesund, Svalbard. One year of lidar data are analysis with respect to their backscatter, depolarisation, and colour ratio, i.e. 475/481 quality assured hours of data, including the vertical distribution and seasonality. In addition, FRP was used to describe the long-term trend of Russian/Canadian Forest fires, meteorological radio-soundings, and model data for the determination of the tropopause altitude. Particle size estimates were presented based on Mie calculations. The origin of the air-masses was calculated using CLaMS simulations, and for two cases back trajectory ensembles were presented, which nicely illustrate the source regions and transport patterns. Thus, it is an interesting publication, and a valuable contribution to the journal, and the science to better understanding the lower Arctic stratospheric aerosol in the Arctic.

I find the article easy to read, well structured, and it is easy to follow the argumentation. Nevertheless, I have a few general comments, which I find should be addressed by the authors. In addition, a number of more specific comments are given below.

General comments:

L 82ff: Description of the article structure, give 3.3., 3.5, 4, 5

   … please complement with description of Section 2, Section 3

L205 ff which data have you used, please give a reference to the FRP dataset, MODIS acquires data at 3 spatial resolution, 250 m (B1-2), 500 m (3-7) and 1 km (8-34), so the gridded data set is not equal to the spatial resolution of the observations

Fig. 3 Using MODIS data for "trend» studies is not as easy, because you will need to account for flaring, industrial pollution (pot. volcanic signals) which also show heat-signatures, account for cloud-cover, and oversampling of off-nadir pixels. Therefore, it might be more useful to use assimilated data (e.g. GFAS) for this. Alternatively, there should be a disclaimer pointing this out clearly. In addition, it might be easier to see pot. trends if one would plot averaged data for the fire-seasons only (e.g., May/April-Sept/Oct.).

Compare also with: Zhu et al, 2021 https://doi.org/10.1029/2021GL096095 or Bondur et al., 2023, https://doi.org/10.3390/fire6030099.

It would be good to make the distinction between what you consider as forest fires and biomass burning a bit clearer throughout the manuscript. It is understandable, but as forest fires are "biomass" burning, it can lead so a bit confusion for the reader. Thus, an additional sentence could help to prevent this.

Specific comments:

L19/20 Stratospheric aerosol can either be long-range transported or even directly created in the stratosphere and has a high sulfuric content
   … maybe have long-range transported/injected …. In contrast to directly created in ….

L32/33 The absence of water vapour and removal by precipitation does not take place in the stratosphere
   … this sentence is unclear. Maybe: Due to the absence …

L40 This is comparable with all volcanic eruptions from 2000 to 2015 (Yu et al., 2017).
   … could you add how much this is compared to Pinatubo, just for the curious reader…

L54 by Vernier et al. (2011) who used satellite data by CALIPSO observations

   …who used observations by Cloud-Aerosol Lidar and Infrared Pathfinder Satellite Observations (CALIPSO)

L 50 As Notholt et al. (2005) show, the sulfur … are constantly decreasing
   … can you find a newer publication (I don't think so, but it could have changed in 20 years)

L65 This observation is becoming more frequent in recent years (Zielinski et al., 2020)
   … not really a "trend paper", maybe e.g. recent observations from Zielinski…
L78/80 It is therefore important to understand the … It is therefore important to understand the
   … to improve the style, I would like to propose to rephrase one of the sentences

L85   and we have a qualitatively good Lidar data set for every month
   … for every month of the year

L105 ff LR
   …. can you give a reference for this choice, what do these values represent?

L121ff: Aren't there no PCSs observed below 27 km? Should be easy to filter those out?

L136ff: LR can be calculated
   ….– you have not done this, or … If I understand it correctly you used literature values (L105ff) ?

L150 9173 m ..

... maybe give a standard deviation

L162 to adjacent data
... typo: two adjacent data

L164 finding a good agreement
... maybe finding a good balance

L178 ERA 5
.. approx. which vertical resolution / pressure level of model level data ?

Fig. 4 I would think the individual observations would be more clear (instead of the difference,) as these show the seasonal variations ... but maybe it is better the way it is presented (was just a thought).

L250 has a higher potential temperature than in winter
... in summer than in winter

L255f In the following study we just concentrate on the year 2021, since we have 481 h for 532 nm and 474 h for 355 nm of qualitatively good Lidar measurement time and is available throughout the entire year as well as every month.
... rephrase sentence, In the following we concentrate on... good Lidar measurements...

L259 103 h of measurement time is available
... 103 hours of measurements

L279 of the analysed year
... of the CR in 2021

L285 decreases towards the end of 285 the year in the discussed altitude range of 330 K to 410 K, but
... to 410 K and with decreasing

**Fig. 6 Can the overlapping dots be replaced by small-non-overlapping rectangular symbols ?**

Fig 7/8 I would remove the x-axis "beta_." of the upper panels to make the plots more clean

L316: should be weak 2-5, medium 5-10 (otherwise, because >2 would otherwise also include the data > 5)

Table 3: are the values in Feb and Nov "0" (Feb has only few data, but November), maybe add a comment about them.

Fig 9/16 … would it make sense to add standard deviations to the profiles (if it does not make the plot to messy) ? In addition, it could be helpful to indicate the tropopause altitudes in the plots.

L353f "The red marks, − and +, represent the monthly median and mean respectively. The blue box indicates the 25th and 75th percentiles, while the black lines stand for the 9th and 91th percentiles."
    .. . can be removed from the text, is identical to the Figure caption

Figure 11. Annual cycle of stratospheric aerosol backscatter coefficient $\beta_{532}$ in 360 K and 18 km height.
    … typo:. 380 K

L366 While the values of $\beta_{532}$ are by about a factor of 2 smaller t
    … factor of 1.5 -2

L 375 by a non-absorbing medium takes is described
    … non-absorbing medium is described

Equation 6: describe n, k
L379 effective radius r - Equation 7: describe r

L395 refractive index of biomass burning aerosols
    … do you have a reference for this value as well ?

L462 … model run time
    … what was set as max time ?

L471 one exemplary trajectory
    …. trajectory ensemble

L504 $\delta$ decreases afterwards with increasing height
    …        decreases with increasing heights

L505 depolarisation throughout the entire lower stratosphere is four for 12 UT
        … ?
L562 small, it is probable that they are long enough in the stratosphere.
    … sentence seems incomplete - stratosphere to ….

L595ff Conclusion and Outlook
   … either all with "." or none
L611: coefficient very well fit to the physical
   …. fit very well

L621/622: A regional radiative forcing of $-0.1$ W/m$_2$ caused by the ATAL was calculated from CALIOP measurements …
   … where is this coming from – reference or your work / I don't see CALIOP measurements in the actual manuscript

L625 volcanic eruptions since 2000. argue that
   … there seems to be a reference missing

L630 ff any recommendation for in-particular-under sampled areas, which should be prioritised? What about CALIOP/EarthCARE .. should they be mentioned?

References: need to be checked, sometimes abbreviations for journals, sometimes full names, partly with small letters, doi is given for parts of the references, for others not, … CO 2 (L 810), …

Is the following paper of relevance?
Yan, X., https://doi.org/10.5194/egusphere-2024-782, 2024.

---

## Author Comment (AC2)

Dear Kerstin Stebel,

Thank you very much for your review. Please find our responses within your notes and comments.

The manuscript by Graβl et al, with Title "Does the Asian Summer Monsoon Play a Role in the Stratospheric Aerosol Budget of the Arctic?" analyses the contribution of Asian Tropopause Aerosol Layer during the Asian summer monsoon to the lower polar stratosphere about Ny-Ålesund, Svalbard. One year of lidar data are analysis with respect to their backscatter, depolarisation, and colour ratio, i.e. 475/481 quality assured hours of data, including the vertical distribution and seasonality. In addition, FRP was used to describe the long-term trend of Russian/Canadian Forest fires, meteorological radio-soundings, and model data for the determination of the tropopause altitude. Particle size estimates were presented based on Mie calculations. The origin of the air-masses was calculated using CLaMS simulations, and for two cases back trajectory ensembles were presented, which nicely illustrate the source regions and transport patterns. Thus, it is an interesting publication, and a valuable contribution to the journal, and the science to better understanding the lower Arctic stratospheric aerosol in the Arctic.

I find the article easy to read, well structured, and it is easy to follow the argumentation. Nevertheless, I have a few general comments, which I find should be addressed by the authors. In addition, a number of more specific comments are given below.

General comments:

L 82ff: Description of the article structure, give 3.3., 3.5, 4, 5

    … please complement with description of Section 2, Section 3
    => thank you. The other sections are added

L205 ff which data have you used, please give a reference to the FRP dataset, MODIS acquires data at 3 spatial resolution, 250 m (B1-2), 500 m (3-7) and 1 km (8-34), so the gridded data set is not equal to the spatial resolution of the observations
    => Thank you for pointing it out. We added the information about the spacial (gridded) resolution of 1km

Fig. 3 Using MODIS data for "trend» studies is not as easy, because you will need to account for flaring, industrial pollution (pot. volcanic signals) which also show heat-signatures, account for cloud-cover, and oversampling of off-nadir pixels. Therefore, it might be more useful to use assimilated data (e.g. GFAS) for this. Alternatively, there should be a disclaimer pointing this out clearly. In addition, it might be easier to see pot. trends if one would plot averaged data for the fire-seasons only (e.g., May/April-Sept/Oct.).

    => Thank you for this remark. You are right that the product behind Fig 3 must not be the best. However, we only use this MODIS product to explain that (as we also see in our lidar data set) there is no indication of high biomass burning in 2021 compared to other years. We assume that in first order all the effects which you describe are more or less similar for different years.

Hence, we use the data just to verify that in 2021 we had a stratosphere which was not affected by pronounced aerosol events and that, therefore, a possible annual cycle can be more easily detected.
We add in the manuscript: "We use this FRP only qualitatively to estimate the relative importance of biomass burning for the year 2021."

Compare also with: Zhu et al, 2021 https://doi.org/10.1029/2021GL096095 or Bondur et al., 2023, https://doi.org/10.3390/fire6030099.

It would be good to make the distinction between what you consider as forest fires and biomass burning a bit clearer throughout the manuscript. It is understandable, but as forest fires are "biomass" burning, it can lead so a bit confusion for the reader. Thus, an additional sentence could help to prevent this.

=> Thanks for this comment. We have cancelled forest fire in the manuscript and will only speak about biomass burning (aerosol).

Specific comments:

L19/20 Stratospheric aerosol can either be long-range transported or even directly created in the stratosphere and has a high sulphuric content
… maybe have long-range transported/injected …. In contrast to directly created in ….
=> changed

L32/33 The absence of water vapour and removal by precipitation does not take place in the stratosphere
… this sentence is unclear. Maybe: Due to the absence …
=> Thank you, it is changed

L40 This is comparable with all volcanic eruptions from 2000 to 2015 (Yu et al., 2017).
… could you add how much this is compared to Pinatubo, just for the curious reader…

=> information added

L54 by Vernier et al. (2011) who used satellite data by CALIPSO observations

…who used observations by Cloud-Aerosol Lidar and Infrared Pathfinder Satellite Observations (CALIPSO)
=> Thank you, changed

L 50 As Notholt et al. (2005) show, the sulfur … are constantly decreasing
         … can you find a newer publication (I don't think so, but it could have changed in 20 years)
         => found a more recent reference

L65 This observation is becoming more frequent in recent years (Zielinski et al., 2020)
         … not really a "trend paper", maybe e.g. recent observations from Zielinski…
         => Thank you, changed

L78/80 It is therefore important to understand the … It is therefore important to understand the
         … to improve the style, I would like to propose to rephrase one of the sentences
         => changed

L85      and we have a qualitatively good Lidar data set for every month
         … for every month of the year
         => changed

L105 ff LR
         …. can you give a reference for this choice, what do these values represent?
         => information added

L121ff: Aren't there no PCSs observed below 27 km? Should be easy to filter those out?
         => After the adjustment of the fitting range we did not find any PSCs in our data set.

L136ff: LR can be calculated
         ….– you have not done this, or … If I understand it correctly you used literature values (L105ff)?

         => Thank you for this comment. In fact, we "played around" a bit with different LRs. You can estimate the stratospheric LR by the constrains: a) (tropospheric) cirrus clouds should have the same backscatter in different wavelengths b) clear altitudes in the free troposphere should have a constant color ratio. Thereby, we found out that the stratospheric LR is larger for 355nm than for 532nm. We will add this information in the new version of the manuscript. Quotes from literature are not very helpful as they mostly deal with strong events (volcanic eruptions or biomass burning). In theses cases generally a slightly higher LR (at 532nm) has been found (but this is not what we are describing in our work).
We add: … if not stated otherwise. "The fact that we found LR532 < LR355 in the stratosphere was derived by our lidar data via two conditions: i) same backscatter at different wavelengths for (tropospheric) cirrus clouds ii) the same gradients for the aerosol backscatter in clear layers of the troposphere."

L150 9137 m
         … maybe give a standard deviation
         => standard deviation added

L162 to adjacent data
    … typo: two adjacent data
    => Changed

L164 finding a good agreement
    … maybe finding a good balance
    => Changed

L178 ERA 5
    .. approx. which vertical resolution / pressure level of model level data?
    => We use ERA5 data on 137 vertical model levels up to 0.01 hPa (~80km), but in lower horizontal and time resolution referred to as ERA5 1° × 1° (similar to Ploeger et al., 2021; Konopka et al., 2022; Clemens et al., 2023). Here, ERA5 data are truncated to a 1◦ × 1◦ horizontal grid and a6-hourly time resolution, whereby the vertical resolution is the same as in the original ERA5 reanalysis (~0.3 to 0.4 km up to 20km).

Fig. 4 I would think the individual observations would be more clear (instead of the difference,) as these show the seasonal variations … but maybe it is better the way it is presented (was just a thought).
    => The plot showing the absolute values only shows the seasonal cycle of the changes in the tropopause in  the radio sonde data as well as in the ERA5 data. Figure 4b shows that the deviations between radio sonde  and ERA5 are about the same throughout the year.

L250 has a higher potential temperature than in winter
    … in summer than in winter
    => changed

L255f In the following study we just concentrate on the year 2021, since we have 481 h for 532 nm and 474 h for 355 nm of qualitatively good Lidar measurement time and is available throughout the entire year as well as every month.
    … rephrase sentence, In the following we concentrate on… good Lidar measurements…
    => changed

L259 103 h of measurement time is available
    … 103 hours of measurements
    => changed

L279 of the analysed year
    … of the CR in 2021
    => changed

L285 decreases towards the end of the year in the discussed altitude range of 330 K to 410 K, but

      … to 410 K and with decreasing

      => Sentence changed

Fig. 6 Can the overlapping dots be replaced by small-non-overlapping rectangular symbols?

      => we already tried this out but then the dots have to be such small, that they are barely visible in the plot. Therefore we decided to let the plots overlap, since it is also just an overview plot, where the exact values at a certain altitude are not crucial for the further investigation and analysis

Fig 7/8 I would remove the x-axis "beta_." of the upper panels to make the plots more clean

      => Thank you for the hint. Both Figures are updated

L316: should be weak 2-5, medium 5-10 (otherwise, because >2 would otherwise also include the data > 5)

      => It is clarified now in the text

Table 3: are the values in Feb and Nov "0" (Feb has only few data, but November), maybe add a comment about them.

      => a short sentence is added

Fig 9/16 … would it make sense to add standard deviations to the profiles (if it does not make the plot to messy)? In addition, it could be helpful to indicate the tropopause altitudes in the plots.

      => Especially in Figure 16 the plot looks very busy with an added standard deviation for all of the lines. In Figure 9 the tropopause is marked for each month.

L353f "The red marks, − and +, represent the monthly median and mean respectively. The blue box indicates the 25th and 75th percentiles, while the black lines stand for the 9th and 91th percentiles."

      ...can be removed from the text, is identical to the Figure caption

      => sentence deleted

Figure 11. Annual cycle of stratospheric aerosol backscatter coefficient $\beta 532$ in 360 K and 18 km height.

      … typo:. 380 K

      => changed

L366 While the values of $\beta 532$ are by about a factor of 2 smaller t

      … factor of 1.5 -2

      => changed

L 375 by a non-absorbing medium takes is described
… non-absorbing medium is described
=> word deleted

Equation 6: describe n, k
L379 effective radius r - Equation 7: describe r
=> parameters got description

L395 refractive index of biomass burning aerosols
… do you have a reference for this value as well?
=> reference added

L462 … model run time
… what was set as max time?
=> All trajectories ended on 1$^{st}$ May 2021 and have therefore a different length

L471 one exemplary trajectory
…. trajectory ensemble
=> changed

L504 δ decreases afterwards with increasing height
…        decreases with increasing heights
=> changed

L505 depolarisation throughout the entire lower stratosphere is four for 12 UT
…?
=> sentence clarified

L562 small, it is probable that they are long enough in the stratosphere.
… sentence seems incomplete -- stratosphere to ….
=> sentence clarified

L595ff Conclusion and Outlook
…  either all with "." or none
=> change

L611: coefficient very well fit to the physical
…. fit very well
=> changed

L621/622: A regional radiative forcing of −0.1 W/m2 caused by the ATAL was calculated from CALIOP measurements …

> … where is this coming from – reference or your work / I don't see CALIOP measurements in the actual manuscript
>> => paragraph is re-written

L625 volcanic eruptions since 2000. argue that
> … there seems to be a reference missing
>> => added

**L630 ff any recommendation for in-particular-under sampled areas, which should be prioritised?**
The tropical western pacific is generally an important region for the intrusion of tropospheric air into the stratosphere (Müller et al 2024 https://acp.copernicus.org/articles/24/2169/2024/). Observations in that region could easily track the possible ascent of aerosol from the Asian monsoon region into the stratosphere. Further, Lidar observations on the southern hemisphere are sparse. Satellites like the upcoming EarthCARE may potentially track the propagation of stratospheric aerosol layers towards the South Pole.

We add „Especially the upcoming EarthCARE satellite may help to track faint stratospheric aerosol layers from the Asian monsoon region also over the Southern hemisphere."

**What about CALIOP/EarthCARE... should they be mentioned?**
> => Thanks for this comment. We will now mention EarthCARE as a valuable future tool in the outlook!

**References: need to be checked, sometimes abbreviations for journals, sometimes full names, partly with small letters, doi is given for parts of the references, for others not, … CO 2 (L 810), …**
> => DOIs are added for every publication

**Is the following paper of relevance?**
**Yan, X., https://doi.org/10.5194/egusphere-2024-782, 2024**

> => Thank you for pointing out this study. We included the findings of Yan et. al. Into ours

---

## Author Response (AR1)

Dear Stelios Kazadzis, dear Reviewers,

Thank you very much for your numerous comments and suggestions to improve the manuscript. Our comments to all of your recommendations can be found in the public discussion section, where you uploaded your files. We indicated changes in the manuscript by coloring this passage red.

Best regards

Sandra Graßl